# A general highly efficient synthesis of biocompatible rhodamine dyes and probes for live-cell multicolor nanoscopy

Jonas Bucevičius [1], Rūta Gerasimaitė [1], Kamila A. Kiszka[2], Shalini Pradhan[1], Georgij Kostiuk[1], Tanja Koenen[2] & Gražvydas Lukinavičius [1] ✉

The development of live-cell fluorescence nanoscopy is powered by the availability of suitable fluorescent probes. Rhodamines are among the best fluorophores for labeling intracellular structures. Isomeric tuning is a powerful method for optimizing the biocompatibility of rhodamine-containing probes without affecting their spectral properties. An efficient synthesis pathway for 4-carboxyrhodamines is still lacking. We present a facile protecting-group-free 4-carboxyrhodamines' synthesis based on the nucleophilic addition of lithium dicarboxybenzenide to the corresponding xanthone. This approach drastically reduces the number of synthesis steps, expands the achievable structural diversity, increases overall yields and permits gram-scale synthesis of the dyes. We synthesize a wide range of symmetrical and unsymmetrical 4-carboxyrhodamines covering the whole visible spectrum and target them to multiple structures in living cells – microtubules, DNA, actin, mitochondria, lysosomes, Halo-tagged and SNAP-tagged proteins. The enhanced permeability fluorescent probes operate at submicromolar concentrations, allowing high-contrast STED and confocal microscopy of living cells and tissues.

Developments in fluorescence microscopy often rely on fluorescent dyes and probes that are highly photostable, bright, and cover a broad spectral range[1]. Multiple super-resolution imaging techniques have become routine experimental tools and are now driving fundamental knowledge of molecular biology[2,3]. The transition from standard imaging of fixed samples towards living specimens in super-resolution microscopy is empowered by the discovery of compatible and cell-permeable fluorescent dyes[4–7], with silicon-rhodamine being the most recognized[8]. In general, rhodamine class dyes and probes are the most often used fluorescent dyes for the labeling of biological targets in living samples[9–12]. This property stems from the rhodamines' dynamic equilibrium between fluorescent zwitterionic (polar) and colorless nonfluorescent (nonpolar) spirolactone species. The equilibrium is environmentally sensitive and influenced by pH, ion concentration, enzyme activity, local microenvironment polarity, or light[13–15].

The nonpolar spirolactone species are responsible for passive cell membrane permeability. Multiple groups have observed that target binding shifts the equilibrium towards the fluorescent zwitterionic state[6,7,16,17]. This results in so-called fluorogenicity, which allows imaging of living samples without washing off the unbound dye. The development of such dyes has recently attracted much attention, and multiple efforts have been focused on the generation of tuned dyes for self-labeling tags (Halo-tag, SNAP-tag, etc.) that form a covalent bond with the fluorophore[6,7,18–20]. This process is irreversible, and even low cell-membrane permeability could be sufficient for target labeling. In contrast, for reversible binding probes, a certain concentration must be reached and maintained inside the cell to efficiently label the target. Moreover, reversibly binding probes are susceptible to efflux pumps, which tend to reduce the probe concentration inside the cell over time[21]. From the user's perspective, fluorogenic probes that do not

[1]Chromatin Labeling and Imaging group, Department of NanoBiophotonics, Max Planck Institute for Multidisciplinary Sciences, Am Fassberg 11, 37077 Göttingen, Germany. [2]Department of NanoBiophotonics, Max Planck Institute for Multidisciplinary Sciences, Am Fassberg 11, 37077 Göttingen, Germany. ✉e-mail: grazvydas.lukinavicius@mpinat.mpg.de

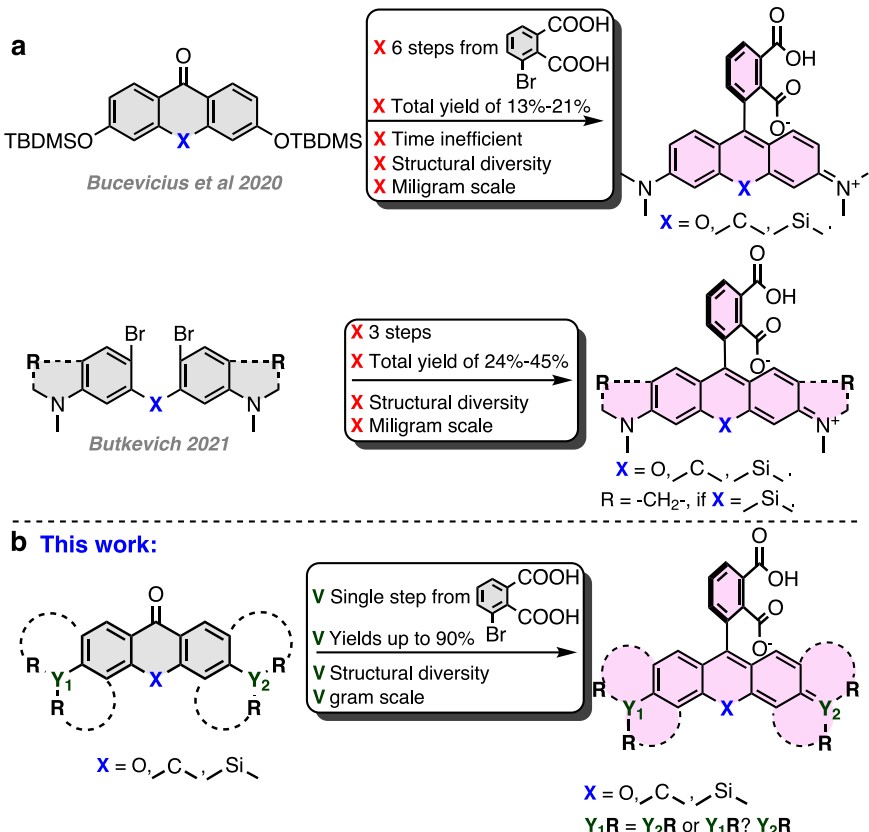

**Fig. 1 | Synthetic approaches to 4-carboxyrhodamines and their analogs. a** Previous work: synthesis of 4-carboxyrhodamines through fluorescein conversion to rhodamine and through bis-aryllanthanum species. **b** This work: synthesis of 4-carboxyrhodamines by single protective group free step.

require genetic manipulations are the most appealing. However, the design of such tools is undoubtedly challenging. Most often, screening studies for the optimal dye–linker–ligand combination are required. Multiple failures are usually due to low cell membrane permeability, off-targeting, a low signal-to-noise ratio, drastically altered lipophilicity, or decreased solubility.

Recently, we discovered that rhodamine 4-isomer-based fluorescent probes, in comparison to isomer-5- or isomer-6-derived probes, demonstrate increased cell permeability and reduced susceptibility to efflux pumps[22]. In isomer-4 probes, the spirolactone forming the carboxyl group is in close proximity to the amide group (or formerly before conjugation–carboxyl), resulting in the phenomenon, which we named the neighboring group effect (NGE). However, the synthesis of such fluorophores required multiple steps and relied on the protection of 3-bromophthalic acid by the formation of bis *tert*-butyl ester. *Tert*-butyl ester mainly creates a steric hindrance around the carbonyl group and does not mask the partial positive charge, resulting in successful lithium–halogen exchange only at low temperatures (−116 °C). At such low temperatures, dialkylamino xanthones turned out to be unreactive; thus, siloxy xanthones had to be employed, followed by five additional synthetic steps[22]. The multistep synthetic path resulted in a relatively low total yield of 13–21% and limited structural diversity (Fig. 1). Recently, Butkevich, A.N. proposed an alternative synthetic strategy relying on the nucleophilic addition of bis-aryllanthanum species to 3-bromophthalic anhydride, followed by palladium-catalyzed carbonylative hydroxylation, resulting in a total 24–45% yield[23] (Fig. 1). While this method did provide 4-carboxy rhodamines in fewer steps and in higher total yield, we found that many starting compounds, especially unsymmetrical dibromides, are challenging or even not possible to obtain.

We aim to further simplify the synthesis of rhodamine 4-isomers by eliminating the need for protecting groups. As a result, herein, we

report a facile, efficient, protecting-group-free, and scalable synthesis of structurally diverse 4-carboxyrhodamines. This allows us to systematically examine the performance of fluorescent probes derived from symmetrical and unsymmetrical fluorophores in living specimens. We exploit the dyes' structural diversity for the selection of highly biocompatible probes and apply them for multicolor confocal microscopy and STED nanoscopy of living cells and tissues. The high cell membrane permeability of the synthesized dyes allows target staining at low nanomolar concentrations with superior specificity, whereas spectral tuning allows positioning the emission maximum of the dye closer to the depletion laser wavelength (commonly used 775 nm) resulting in reduced $I_{sat}$ values. Our results clearly demonstrate the high potential of rhodamine 4-isomers for the development of biocompatible fluorescent probes.

## Results
### Synthesis of 4-carboxyrhodamine dyes

In situ formed carboxylic acid salts may be applied to suppress carbonyl group reactivity towards nucleophiles[24], but polar carboxylate salts tend to have low solubility in ethereal solvents, which are typically used in lithium–halogen exchange reactions. We prepared and investigated the solubility of 3-bromophthalic acid salts in anhydrous tetrahydrofuran (THF) and diethyl ether (Et₂O). Neither disodium nor dilithium 3-bromophthalates were soluble. Monosodium salt showed little solubility, while monolithium salt was sufficiently soluble in THF at ambient temperature. Lithium hydrogen phthalate stands out from other phthalate salts with one of the shortest intramolecular hydrogen bonds observed (reported O–O distance 2.385 Å) and a planar phthalate ion structure[25], which both are strong indications of significantly reduced acidity, especially in organic solvents. This encouraged us to perform lithium–halogen exchange with lithium hydrogen 3-bromophthalate (Table 1, **I**) under standard conditions followed by

**Table 1 | Optimization of the Li-halogen exchange reaction conditions of 3-bromophthalic acid**

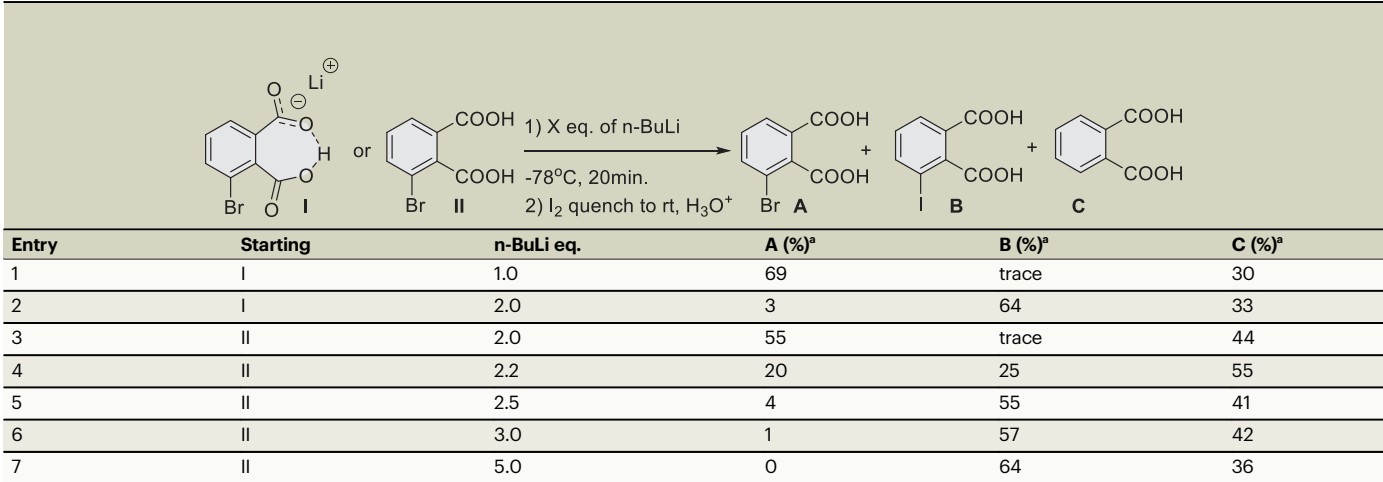

| Entry | Starting | n-BuLi eq. | A (%)[a] | B (%)[a] | C (%)[a] |
|---|---|---|---|---|---|
| 1 | I | 1.0 | 69 | trace | 30 |
| 2 | I | 2.0 | 3 | 64 | 33 |
| 3 | II | 2.0 | 55 | trace | 44 |
| 4 | II | 2.2 | 20 | 25 | 55 |
| 5 | II | 2.5 | 4 | 55 | 41 |
| 6 | II | 3.0 | 1 | 57 | 42 |
| 7 | II | 5.0 | 0 | 64 | 36 |

Data in the table represent the mean ($N = 2$).
[a]Conversion was determined by LC/MS analysis.

**Table 2 | Optimization of nucleophilic addition reaction conditions to silaxanthone**

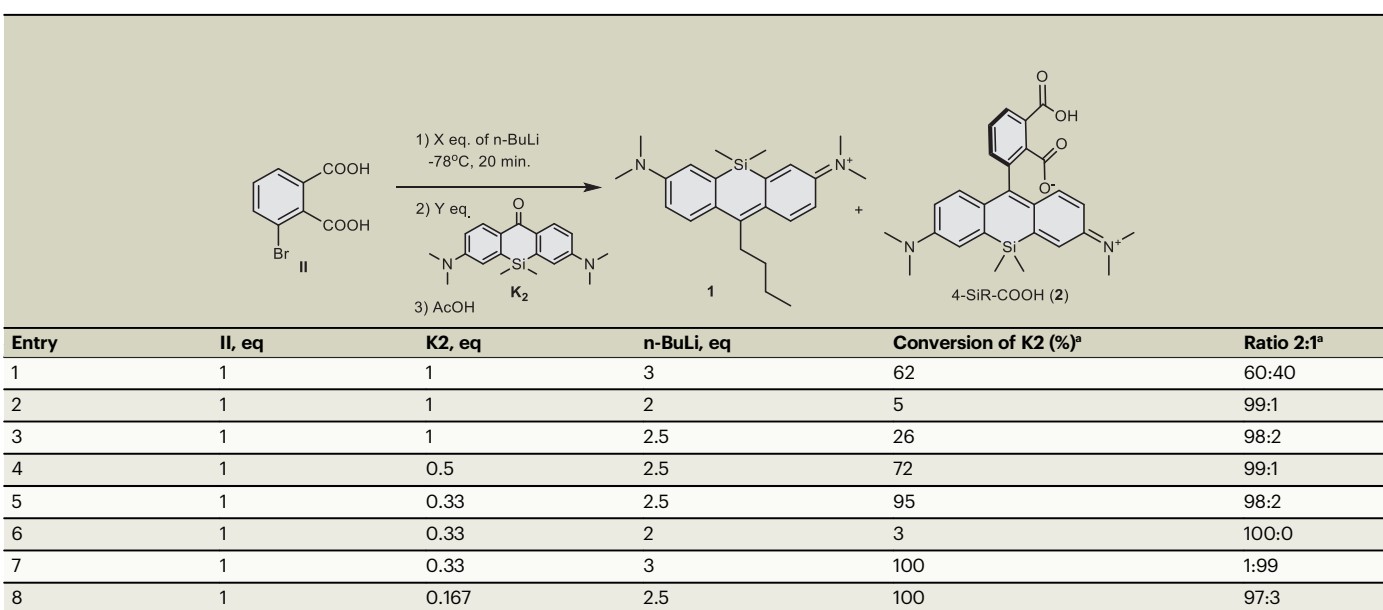

| Entry | II, eq | K2, eq | n-BuLi, eq | Conversion of K2 (%)[a] | Ratio 2:1[a] |
|---|---|---|---|---|---|
| 1 | 1 | 1 | 3 | 62 | 60:40 |
| 2 | 1 | 1 | 2 | 5 | 99:1 |
| 3 | 1 | 1 | 2.5 | 26 | 98:2 |
| 4 | 1 | 0.5 | 2.5 | 72 | 99:1 |
| 5 | 1 | 0.33 | 2.5 | 95 | 98:2 |
| 6 | 1 | 0.33 | 2 | 3 | 100:0 |
| 7 | 1 | 0.33 | 3 | 100 | 1:99 |
| 8 | 1 | 0.167 | 2.5 | 100 | 97:3 |

Data in the table represent the mean ($N = 2$).
[a]Conversion and ratio were determined by LC/MS analysis.

quenching with excess $I_2$ and LC/MS analysis. Upon the addition of 1 eq of n-BuLi, no significant dilithium salt precipitation was observed, and the solution turned light yellow, indicating some formation of ArLi species. LC/MS analysis revealed that the obtained mixture consisted of unreacted starting bromide, phthalic acid, and a trace amount of 3-iodophthalic acid (Table 1, entry 1). Increasing n-BuLi to 2 eq resulted in 64% conversion to 3-iodophthalic acid (Table 1, entry 2). After the first successful attempts, we switched the starting material from lithium hydrogen 3-bromophthalate (Table 1, **I**) to 3-bromophthalic acid (Table 1, **II**), as the monolithium salt can be formed in situ by neutralization of 3-bromophthalic acid with an additional equivalent of n-BuLi. Thus, the addition of 2 eq of n-BuLi resulted in only trace formation of 3-iodophthalate (Table 1, entry 3), but 55–64% conversion to 3-iodophthalate was achieved once 2.5, 3, or 5 eq of n-BuLi were used (Table 1, entries 5–7). This indicates that ≥2.5 eq is the optimal amount of BuLi for the efficient formation of ArLi species. Inspired by these

results, we decided to further examine the nucleophilic addition of the formed ArLi species to silaxanthone **K2** (Table 2), which should afford the 4-SiR-COOH (**2**) fluorophore in a single step without any protecting groups or any need for functional group interconversions. First, we used a stoichiometric ratio of reactants: 1 eq of 3-bromophthalic acid and 3 eq of n-BuLi followed by the addition of 1 eq of ketone (**K2**). The obtained mixture was analyzed with LC/MS, indicating 62% conversion of ketone, but two products were formed: target compound **2** and n-BuLi addition product **1** with a ratio of 6:4 (Table 2, entry 1). The formation of **1** indicated that unreacted n-BuLi was still present in the reaction mixture. Reduction of the n-BuLi amount to 2 eq resulted in decreased conversion of **K2** to 5% and a trace formation of side product **1** (Table 2, entry 2). The use of 2.5 eq resulted in increased conversion to 26% and retained the selectivity ratio (Table 2, entry 3), indicating an optimal n-BuLi quantity. Abatement of ketone to

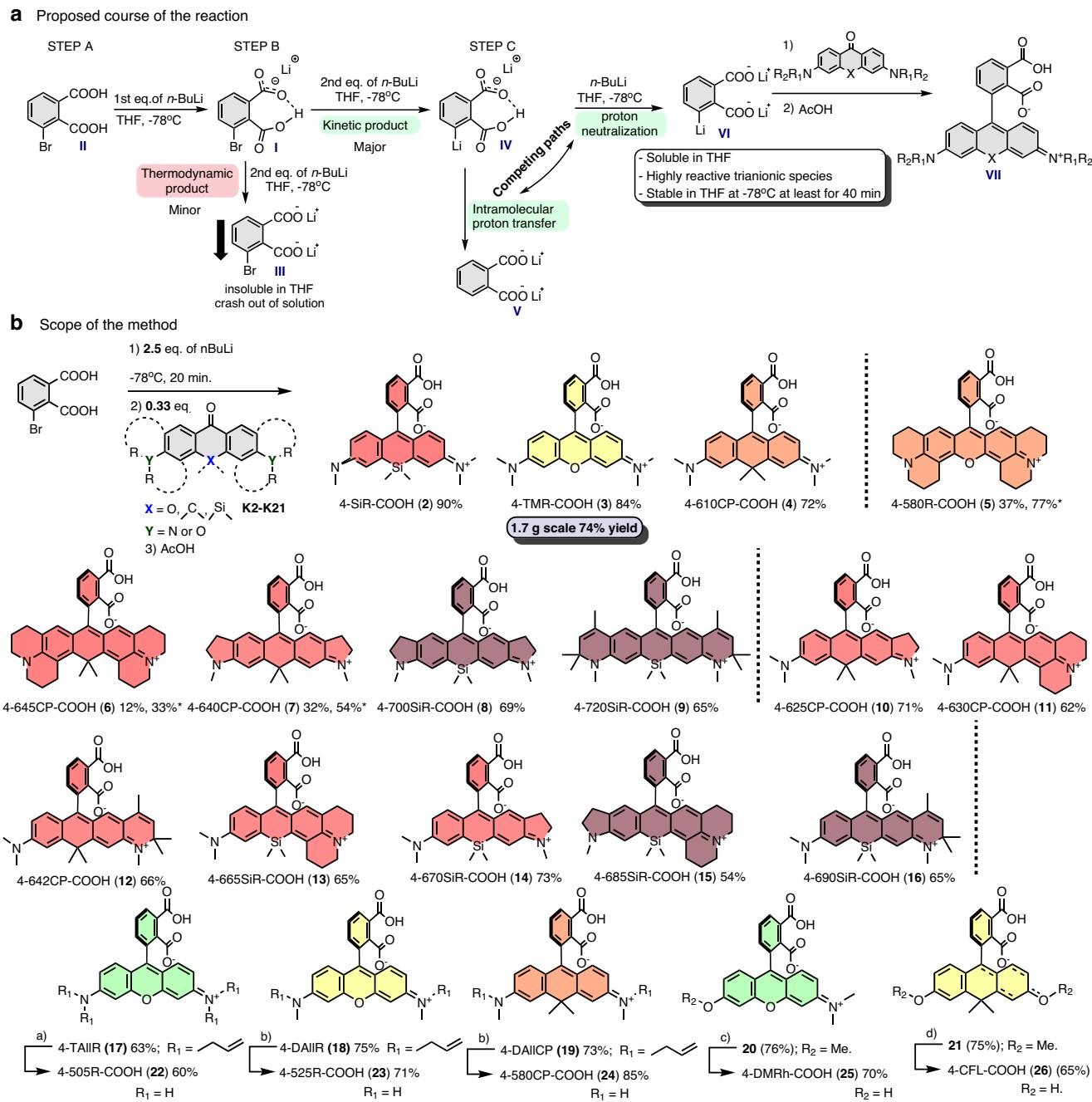

**Fig. 2 | Reaction course and scope. a** Proposed course of the lithium halogen exchange reaction of 3-bromophthalic acid. **b** The scope of the described 4-carboxyrhodamine synthetic method (a) 14 mol% Pd(PPh₃)₄, 7 eq NDMBA, MeCN. 45 °C, 10 h. (b) 7 mol%, Pd(PPh₃)₄, 2.5–3 eq NDMBA, MeCN, 40 °C, overnight. (c) 40 eq BBr₃, 1,2-DCE, 55 °C, 24 h. (d) 40 eq BBr₃, 1,2-DCE, 55 °C, 72 h. *Marks the yields when 0.167 eq (instead of 0.33 eq) of ketone reactant was used.

0.5 eq and eventually to 0.33 or 0.167 eq led to optimal conditions with 95–100% conversions of **K2** and with an ~98:2 selectivity ratio (Table 2, entries 4–5 and 8), resulting in a high 90% yield of 4-SiRCOOH (**2**) after flash column purification (Table 2, entry 5).

Relying on the data obtained in the optimization of the experimental conditions, we propose the following reaction course (Fig. 2a). In STEP A, lithium 3-bromophthalate (**I**) is formed after the addition of 1 eq *n*-BuLi to 3-bromophthalic acid (**II**). In STEP B, due to the significantly reduced acidity of **I**, the reaction can follow thermodynamic or kinetic paths, forming either dilithium salt **III** or ArLi species **IV**, respectively. In our favor, the thermodynamic product was not dominant even at −78 °C, and the reaction mainly followed a kinetic

pathway yielding **IV**. Once formed, ArLi species **IV** are metastable and may self-quench by intramolecular proton transfer, forming dilithium phthalate **V**, or the proton can be neutralized by additional *n*-BuLi and form highly reactive trianionic ArLi species **VI**. These reaction paths are competing, which results in the nonstoichiometric requirement of *n*-BuLi. We did not observe significant degradation of **VI** in THF at −78 °C for at least 40 and 20 min of aging was sufficient. Due to multiple reaction paths, **VI** forms from starting compound **II** nonstoichiometrically, and a reduced ratio of electrophilic reactive partner (xanthone) must be used (Fig. 2a).

To demonstrate the scope of the developed synthesis method, we focused on the preparation of a series of structurally diverse xanthones **K2**–**K21** (Supplementary Fig. 1). Xanthones **K2**–**K4** were

synthesized as previously published[8,26,27]. Xanthones with a carbon bridging atom (**K6**, **K7**, and **K10–K12**) were synthesized by Lewis acid-mediated condensation of substituted benzyl alcohols and substituted α-methylstyrenes, followed by Brønsted acid-initiated cyclization and finally oxidation with $KMnO_4$ (Supplementary Fig. 2). Symmetrical silaxanthones (**K8**, **K9**) were obtained by condensation of two brominated aniline-type units with formaldehyde followed by dilithiation and substitution with $Me_2SiCl_2$ to introduce a bridging fragment and finally temperature-controlled oxidation with $KMnO_4$ (Supplementary Fig. 3a). For unsymmetrical silaxanthones (**K13–K16**), the first step was performed by $BF_3\text{-}OEt_2$-mediated condensation[28] of bromo-substituted benzylic alcohols with the 3-bromoaniline-type counter partner, followed by the same synthetic steps as for symmetrical ketones (Supplementary Fig. 3b). Xanthones with methoxy and allyl amino substituents (**K17–K21**) were obtained by alkylation of hydroxyxanthones or by nucleophilic aromatic substitution of 3,6-bistriflate xanthones with corresponding amines (Supplementary Fig. 4). With the key intermediates **K2–K21** in hand, we explored the scope of the developed 4-carboxyrhodamine synthesis method (Fig. 2b). Classical oxygen-, carbon- or silicon-bridged rhodamines (**2–4**) were obtained in significantly fewer synthetic steps and 72–90% yields, which is 3–7-fold higher than the previously published synthetic procedures[22,23] (Fig. 2b). In addition, the synthetic method was found to be compatible with ketones possessing amino groups fused to the aromatic ring (Supplementary Fig. 1, **K5–K19**), providing compounds **5–9** in 12–77% yields. The reduced yields for **5–7** can be attributed to the reduced solubility of highly rigid symmetrical ketones **K5–K7**. They tend to crash out of the mixture, resulting in heterogeneous conditions, thus prolonging the reaction time, which are incompatible with the stability of ArLi species **VI**. Increasing 3-bromophthalic acid to 6 eq (abatement of ketone to 0.167 eq) recovered the reaction yields (Fig. 2b, yields marked with *). Unsymmetrical ketones **K10–K16** showed good solubility in THF, resulting in usually high 54–71% yields for compounds **10–16**. In addition, rhodamines possessing primary (**22**) or secondary amino (**23**, **24**) substituents were obtained via palladium-catalyzed allyl group cleavage of the compounds with allyl amino substituents **17–19**. Similarly, rhodol (**25**) or carbofluorescein (**26**) were obtained from methoxy-protected intermediates **20**, **21** after $BBr_3$-mediated demethylation (Fig. 2b).

Finally, we demonstrated the scalability of the developed synthetic method by performing a gram-scale synthesis of the 4-TMR-COOH (**3**) dye. We obtained 1.7 g of product from a single batch in 74% yield after flash column purification without any scrutinized scale-up optimizations. In summary, the described facile, highly efficient and scalable synthetic method provides access to previously inaccessible structurally diverse 4-carboxyrhodamines and their analogs with a wide range of photophysical properties.

## Synthesis of fluorescent probes

Rhodamine dyes with a pendant carboxyl group can be attached to small molecule ligands/inhibitors to provide fluorescent probes with high specificity in cellular staining[9]. We evaluated the performance of isomer-4 dyes by conjugating them via peptide coupling to the range of targeting moieties (Fig. 3) and obtaining a library of fluorescent probes targeting SNAP-tag (**27–46, 105–109**), β-tubulin (**47–67**), Halo-tag (**68–79**), mitochondria (**80–86**), DNA (**87–92**), lysosomes (**93–99**), and actin (**100–104**).

## Characterization of the fluorescent dyes and probes in vitro

The synthesized dyes were characterized by means of absorption and fluorescence spectroscopy, fluorescence quantum yield, and lifetime measurements in phosphate-buffered saline (PBS) (Table 3). Additionally, we have estimated hydrophobicity of these dyes by calculating $c$Log$P$ values of spirolactone and

zwitterion states (Table 3). Furthermore, photophysical properties were assessed with 0.1% sodium dodecyl sulfate additive to characterize the dyes, avoiding any possible dye aggregation effects (Supplementary Table 1). The absorption maxima of the obtained dyes are in the range from 500 to 720 nm, and the corresponding emission maxima are in the range from 522 to 757 nm. This coverage is achieved via the introduction of substituted amino groups fused to the aromatic ring (indoline, julolidine, dihydroquinoline), variation of amino group substitution pattern (tertiary, secondary, primary), alternation of bridging atom (O, C, Si) and desymmetrization of rhodamine core (**10–16**) (Table 3 and Supplementary Table 1).

To characterize the ability of the synthesized fluorophores and probes to switch between spirolactone and zwitterion states, we measured the absorbance in water–1,4-dioxane mixtures with known dielectric constants[29]. We fitted the absorbance change to the dose–response equation and obtained the $^{dye}D_{50}$ value, which indicates the dielectric constant at which absorbance is halved (Fig. 4a, Supplementary Fig. 5, and Supplementary Table 2).

Fluorescent substrates of the self-labeling tags offer much flexibility for the design of imaging experiments. The 4-(benzyloxy)-2-chloropyrimidine (CP) ligand is a structurally simple and commercially available SNAP-tag ligand (Fig. 3). Thus, we synthesized a full range of substrates via peptide coupling (**27–45**) and used this series to compare the $^{dye}D_{50}$ and $^{probe}D_{50}$ values. In all cases, $^{probe}D_{50}$ was significantly increased compared to the corresponding $^{dye}D_{50}$, indicating that for all tested dyes, the conversion of the carboxyl group to amide induces a higher spirolactone content due to NGE[22]. We found that $D_{50}$ values are mainly affected by the bridging atom O < C < Si and, to a lesser extent, by the side substituents. With respect to side substituents, $D_{50}$ increased in the following order: julolidine < indoline < dihydroquinoline ≈ (dimethyl)amino. Interestingly, the spirolactone form was dominant in water for the majority of the silarhodamines, resulting in $D_{50}$ values above or close to 80 (Fig. 4a and Supplementary Table 2), except those possessing the julolidine fragment (4-665SiR (**13**) and 4-685SiR (**15**)).

For this reason, we decided to measure alternative $^{dye}K_{L–Z}$ and $^{probe}K_{L–Z}$ values, which can also be used to characterize the dye's propensity to exist in spirolactone or in zwitterion form[6] (Fig. 4b and Supplementary Table 2). Surprisingly, we did not observe any distinct correlation between $D_{50}$ and $K_{L–Z}$ values. Most likely, both concepts have no direct physical link to the true equilibrium constants and are just empirical methods, which provide a numerical value for a basic comparison of the environmental sensitivity of rhodamine dye. We noticed that the concept of $K_{L–Z}$ proved to be more suitable to assess dyes that are highly closed in a water environment but failed to give comparative values for the dyes that mainly exist in zwitterion form in 1:1 water–dioxane mixtures ($\varepsilon = 35$). Conversely, the $D_{50}$ concept proved to be more suitable to assess rhodamine dyes, which preferentially exist as zwitterions in water or dyes that show little sensitivity to changes in the surrounding media.

4-CFL (**26**) dye belongs to the fluorescein dye class and did not demonstrate a distinct response to water content in 1,4-dioxane. It is known that fluoresceins are highly sensitive to pH. We measured the pH dependence of the absorption of the 4-CFL-COOH (**26**) dye and found that the dye partially exists in an open-fluorescent form in the physiological pH range and is suitable for live cell microscopy (Supplementary Fig. 6).

## Characterization of the fluorescent dyes and probes in silico

We performed density functional theory (DFT) calculations using the Gaussian 09 program package[30] to estimate the influence of substituents on the rhodamine core on the absorption properties and the impact on the energies of frontier molecular orbitals (FMOs). The geometries of the studied compounds were

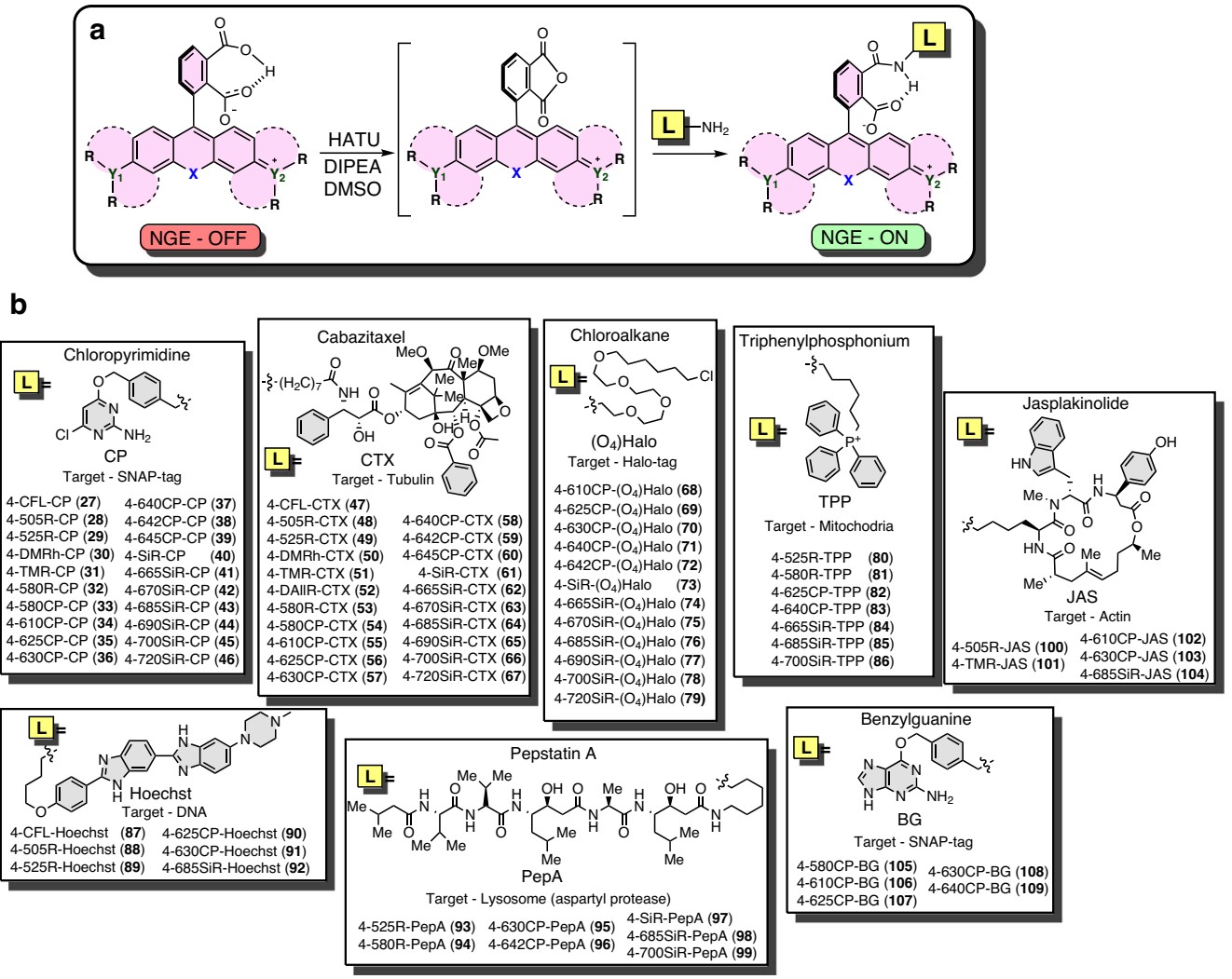

**Fig. 3 | Synthesis of a library of STED imaging-compatible fluorescent probes targeting different intracellular targets. a** General conjugation reaction scheme. **b** Structures of the synthesized conjugates.

optimized using the Becke, 3-parameter, Lee–Yang–Parr (B3LYP) exchange–correlation hybrid functional together with the 6-311++G(d,p) basis set, and water was implemented with the integral equation formalism for the polarizable continuum model (IEFPCM) based solvation model[31]. We calculated the energies and generated isodensity surface plots of the frontier molecular orbitals. It was found that side substituents and bridging atoms have minimal influence on the lowest unoccupied molecular orbital (LUMO) energies. However, a systematic increase in highest occupied molecular orbital (HOMO) energies was evident in line with the increasing donor strength of the substituents and bridging atom, which resulted in a steadily decreasing HOMO–LUMO gap and a gradual redshift in absorption (Supplementary Fig. 7). The absorption maxima wavelengths in water (IEFPCM) were calculated from the vertical excitations obtained by (time-dependent density-functional theory) TDDFT (time-dependent density-functional theory) calculations[32]. It is known that TDDFT gives a systematic error in the calculation of vertical excitation energy values for dyes with methyne fragments[33]. After applying a systematic empirical correction factor of 0.4 eV, the calculated absorption values were in very good agreement with the experimentally measured values for all studied dyes (Fig. 4c, Supplementary Table 3). The obtained calculation data gave us insight  to include the unsymmetrical rhodamines into the study.

Previously, we compared DFT-calculated free Gibb's energies of isomer 4/5/6-rhodamine dyes and probes upon cyclization of zwitterion to spirolactone in different surrounding environments[22]. However, we overlooked the key reason why the spirolactone-zwitterion equilibrium of 4-isomer rhodamines is drastically affected upon carboxyl group ($^{dye}D_{50}$) interconversion to amide ($^{probe}D_{50}$). To gain better insight into how NGE works, we performed potential energy calculations along the increasing dielectric constant for zwitterion and spirolactone forms of 4-TMR-COOH (**3**) and of a hypothetical model compound 4-TMR-CONHMe. The latter compound served as a model for the probe with a truncated linker and ligand. The obtained energy values of both forms were plotted against the dielectric constant (Fig. 4d). The intersection point of the two curves corresponds to equal potential energies, hence an equal distribution of spirolactone and zwitterion species, and can be regarded as the simulated $D_{50}$ value. The simulated $^{dye}D_{50}$ value was 7, and $^{probe}D_{50}$ was 31, which is in rather good agreement with the experimentally obtained values of 14 and 23, respectively, and an indication that DFT is able to predict NGE behavior. Based on the data obtained from DFT calculations, we hypothesize that the spirolactone form is stabilized by the intramolecular hydrogen bond (Fig. 4e) in both the dye (4-TMR-COOH (**3**)) and the probe (4-TMR-CONHMe) to a considerably similar extent. Hence, the major difference could be attributed to the (de)stabilizing effects of the zwitterion form. Taking into consideration the unprecedented

**Table 3 | Photophysical properties and cLogP values of the synthesized 4-carboxyrhodamines in PBS at pH = 7.4**

| Dye(–COOH) | $\lambda_{max}^{abs}$ (nm) | $\lambda_{max}^{em}$ (nm) | $\varepsilon*10^2$ (m$^{-1}$ cm$^{-1}$) | QY (%) | $\tau$ (ns) | cLogP | |
|---|---|---|---|---|---|---|---|
| | | | | | | Zwitterion | Spirolactone |
| 4-505R (22) | 500 | 522 | 677 ± 16 | 82 ± 3 | 3.99 | −2.2 | 1.5 |
| 4-525R (23) | 522 | 545 | 804 ± 29 | 94 ± 4 | 4.09 | −0.6 | 2.9 |
| 4-DMRh (25) | 521 | 547 | 696 ± 23 | 18 ± 1 | – | −4.3 | 3.4 |
| 4-TAllR (17) | 548 | 570 | 1032 ± 116 | 66 ± 5 | 3.10 | −0.7 | 7.3 |
| 4-DAllR (18) | 549 | 572 | 811 ± 24 | 53 ± 1 | 2.67 | −2.1 | 5.8 |
| 4-CFL (26) | 549 | 574 | 133 ± 1 | 63 ± 3 | 3.55 | −3.3 | 3.6 |
| 4-TMR (3) | 551 | 573 | 776 ± 50 | 41 ± 1 | 2.04 | −3.4 | 4.2 |
| 4-580 R (5) | 579 | 599 | 808 ± 49 | 75 ± 2 | 4.33 | 3.6 | 6.4 |
| 4-580CP (24) | 585 | 609 | 1008 ± 29 | 60 ± 2 | 3.55 | 1.6 | 3.9 |
| 4-610CP (4) | 611 | 636 | 1014 ± 84 | 50 ± 1 | 3.12 | −3.0 | 5.2 |
| 4-DAllCP (19) | 611 | 636 | 975 ± 150 | 59 ± 1 | 3.55 | −1.7 | 6.8 |
| 4-625CP (10) | 621 | 648 | 920 ± 13 | 41 ± 2 | 2.86 | 2.6 | 5.5 |
| 4-630CP (11) | 624 | 644 | 1055 ± 44 | 64 ± 4 | 3.68 | 3.5 | 6.3 |
| 4-640CP (7) | 636 | 661 | 971 ± 61 | 31 ± 1 | 2.44 | 2.9 | 5.8 |
| 4-642CP (12) | 641 | 678 | 1069 ± 23 | 16 ± 1 | 1.27 | 4.5 | 7.4 |
| 4-645CP (6) | 642 | 658 | 1025 ± 80 | 69 ± 3 | 3.83 | 4.6 | 7.4 |
| 4-SiR (2) | 649 | 669 | 342 ± 27 | 43 ± 2 | 2.77 | −0.9 | 5.9 |
| 4-665SiR (13) | 663 | 683 | 978 ± 51 | 39 ± 1 | 2.62 | 4.1 | 7.0 |
| 4-670SiR (14) | 668 | 693 | 455 ± 42 | 28 ± 1 | 1.98 | 3.3 | 6.1 |
| 4-685SiR (15) | 687 | 708 | 873 ± 34 | 30 ± 1 | 2.34 | 4.3 | 7.2 |
| 4-690SiR (16) | 687 | 719 | 265 ± 12 | 10 ± 1 | 0.93 | 5.1 | 8.0 |
| 4-700SiR (8) | 694 | 721 | 475 ± 3 | 17 ± 1 | 1.53 | 3.5 | 6.4 |
| 4-720SiR (9) | 721 | 757 | 184 ± 5 | 8 ± 1 | 0.91 | 7.2 | 10.1 |

Data in the table represent the mean ± SD ($N$ = 3).

short intramolecular hydrogen bond length in 4-TMR-COOH (**3**) (Fig. 4f), it can be considered highly stabilizing, as the proton can be shared by both carboxyl groups, which is also supported by previous observations based on X-ray data of lithium hydrogen phthalate[25]. However, in the case of 4-TMR-CONHMe, the amide hydrogen is not labile and cannot be intershared with the carboxylate group due to the large basicity difference of the conjugate bases. This results in the formation of a standard length (strength) hydrogen bond and a more twisted carboxylate group (Fig. 4f), which destabilizes the zwitterion form and results in an increased $D_{50}$ value of the probe. In summary, an increase in the $^{probe}D_{50}$ value (with respect to $^{dye}D_{50}$) results from the loss of the shared proton stabilizing effect between amide and carboxylate groups in the zwitterion form of the probe.

## Substrates for self-labeling tags

SNAP-tag is a modified human O6-alkylguanine-DNA alkyltransferase (hAGT) accepting CP and O6-benzylguanine (BG) derivatives[34,35]. To comprehensively cover the variety of self-labeling tag substrates, in addition to the synthesized CP probes **27–46**, we conjugated the selected fluorophores to BG to yield compounds **105–109**. A recent study reported that more hydrophobic CP substrates show 4−14-fold slower reaction kinetics than similar BG substrates[36], suggesting that BG substrates might perform better with respect to labeling efficiency. In addition, we also synthesized a series of chloroalkane-PEG$_4$ derivatives **68−79** ((O$_4$)Halo) that react with one of the most frequently used self-labeling tags−Halo-tag. It is based on engineered bacterial dehalogenase (DhaA from *Rhodococcus* sp.), which accepts halogenated alkanes as substrates[37]. Furthermore, we examined the reactivity of all fluorescent substrates by incubating them with a twofold excess of the respective tag protein and detected whether protein labeling took place on a denaturing polyacrylamide gel (SDS−PAGE). Most of the CP-based substrates showed a successful reaction with SNAP-tag protein, though 4-SiR-CP (**40**), 4-665SiR-CP (**41**), 4-670SiR-CP (**42**) substrates showed incomplete reaction and 4-690SiR-CP (**44**), 4-720SiR-CP (**46**) showed no reaction (Supplementary Fig. 8). All of the BG-based

substrates reacted with SNAP-tag protein (Supplementary Fig. 9). Similarly, in the case of Halo-tag substrates, the 4-720SiR-(O$_4$)Halo (**79**) and 4-690SiR-(O$_4$)Halo (**77**) probes showed no or incomplete reaction (Supplementary Fig. 10).

We also determined the fluorogenicity−the increase in fluorescence and absorbance upon reaction with the target protein. All of the SNAP-tag interacting substrates demonstrated absorbance and fluorescence increases upon reacting with SNAP-tag protein, except 4-CFL-CP (**27**). In the CP series, 4-685SiR-CP (**43**) and 4-700SiR-CP (**45**) showed the largest increase, with a more than 14-fold increase in absorbance and a more than 80-fold increase in fluorescence signal (Supplementary Figs. 11 and 12). In the BG series, 4-630CP-BG (**108**) demonstrated an 8-fold increase in absorption and a 6-fold increase in fluorescence signal (Supplementary Fig. 13). We noticed that CP-based SNAP-tag substrates demonstrated higher fluorogenicity in vitro than the corresponding BG series. 4-685SiR-(O$_4$)Halo (**76**) and 4-700SiR-(O$_4$) Halo (**78**) showed the highest (~20-fold) increase in fluorescence signal in the Halo-tag substrate series (Supplementary Fig. 14). In general, the fluorescence increase after reaction with the target protein of the studied self-labeling substrates was in line with the increasing $D_{50}$ value, but the extent of the fluorescence increase was highly dependent on the attached targeting moiety (CP, BG, or (O$_4$)Halo).

Halo and SNAP tags were initially evolved with rhodamine 6-isomer substrates. To test the suitability of the 4-isomer self-labeling tag substrates *in cellulo*, we used engineered U-2 OS cells expressing histone-H3-SNAP (Supplementary Fig. 15), Nup96-SNAP[38] or vimentin-Halo[39,40]. We observed less efficient staining of SNAP-tagged histone H3 (Supplementary Fig. 16) by CP probes compared to BG derivatives[36]. We identified 4-625CP-BG (**107**) and 4-630CP-BG (**108**) as the best-performing substrates. High staining efficiency allowed the acquisition of high-quality STED microscopy images (Supplementary Fig. 17). All Halo-tag substrates, except 4-690SiR-(O$_4$)Halo (**77**) and 4-720SiR-(O$_4$)Halo (**79**), efficiently labeled vimentin-Halo-tag

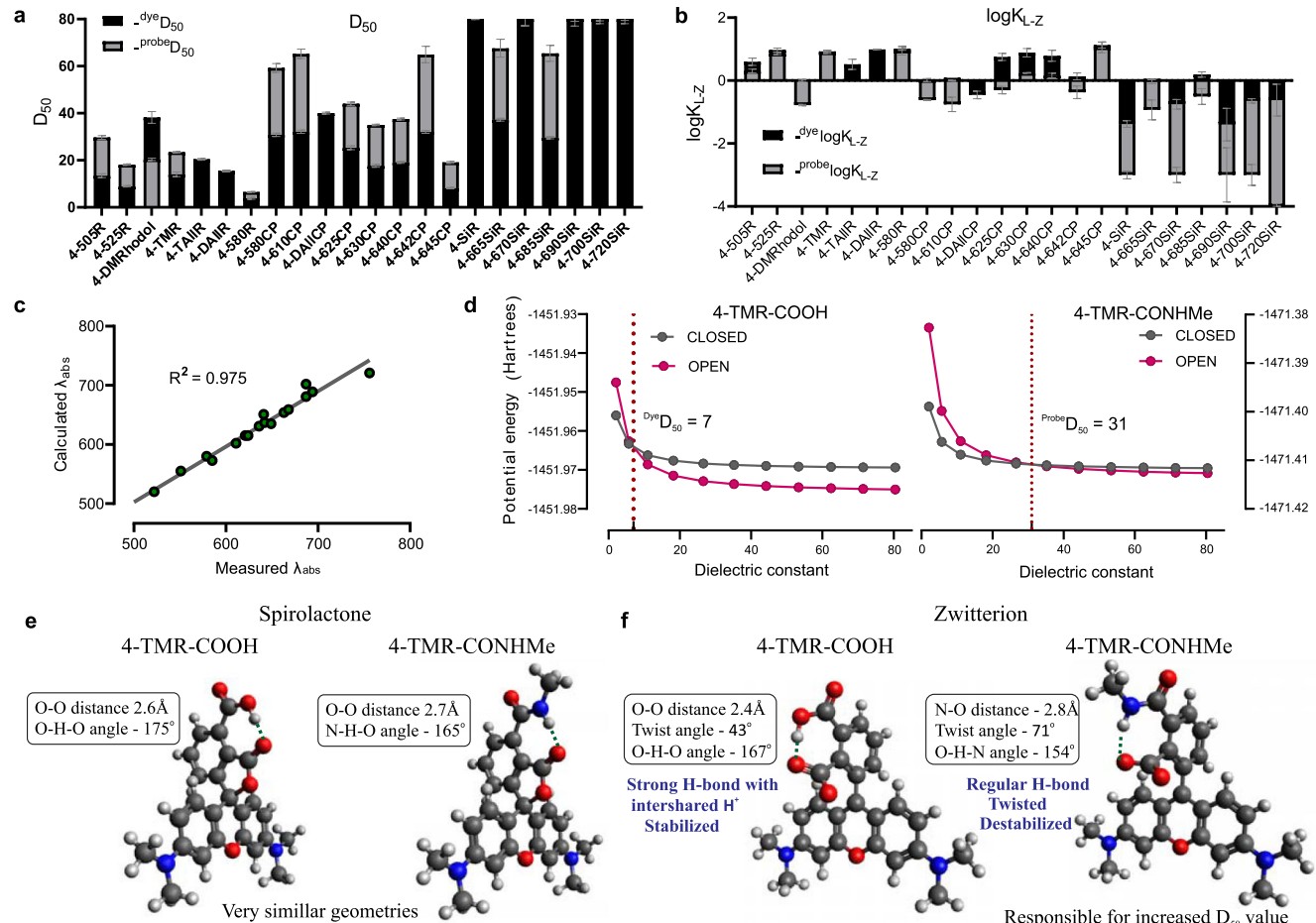

**Fig. 4 | In vitro and in silico characterization of the dyes. a** $^{dye}D_{50}$ and $^{probe}D_{50}$ and **b** $^{dye}\log K_{L-Z}$ and $^{probe}\log K_{L-Z}$ values represented in a stacked bar chart. Data are presented as mean values ± SD, $n = 3$ independent experiments. **c** Correlation between TDDFT calculated and experimentally measured absorption maxima of the dyes in water. **d** Plots of DFT calculated potential energies of zwitterion and spirolactone states against the simulated dielectric constant, intersection point of the curves represent a simulated $D_{50}$ value. **e** DFT B3LYP 6-311 + +G(d,p) IEFPCM optimized geometries of 4-TMR-COOH and 4-TMR-CONHMe spirolactone and **f** zwitterion forms in water. Source data are provided with this paper.

(Supplementary Fig. 18). Thus, in a real experiment, the substrate can be chosen according to photophysical properties, e.g., to ideally match the available excitation laser or to minimize cross-talk in multicolor imaging.

**Tubulin fluorescent probes**

Our previous study showed that the incorporation of rhodamine 4-isomers enhances the cell-membrane permeability of fluorescent probes[22]. We were interested to see whether biocompatibility and staining performance would be on the same level for a wider variety of 4-isomer-based probes. The use of the self-labeling tags for this experiment is suboptimal, as an irreversible reaction with the target makes the staining efficiency less dependent on substrate permeability. Thus, we synthesized a complete series of cabazitaxel (CTX) derivatives targeting tubulin.

We found that almost all probes, except 4-580R-CTX (**53**), 4-690SiR-CTX (**65**), and 4-720SiR-CTX (**67**), were able to stain microtubules in living cells at nanomolar concentrations (Fig. 5a). Indeed, the majority of compounds (>75%) displayed $EC_{50} < 100$ nM, indicating good permeability of the probes (Fig. 5b and Supplementary Fig. 19). The best-performing tubulin probes were 4-625CP-CTX (**56**) and 4-630CP-CTX (**57**), which provided excellent staining at a concentration of 10 nM within 1 h (Fig. 5c and Supplementary Fig. 20). We observed signal-to-background ratios of 20 and 12 in the acquired confocal and 3D STED images of microtubules stained

with 10 and 100 nM 4-625CP-CTX (**56**) or 4-630CP-CTX (**57**), respectively. Only a two-fold signal intensity difference was observed after a 10-fold increase in probe concentration (10 vs. 100 nM, Supplementary Fig. 20). This observation is in good agreement with the low nanomolar $EC_{50}$ values (Fig. 5b, Supplementary Table 4) and demonstrates labeling saturation. It should be noted that for long-term staining, lower concentrations are beneficial because of reduced potential cytotoxic effects. Furthermore, high efflux pump activity in Vero and U-2 OS cell lines did not hamper microtubule staining with 4-625CP-CTX (**56**) and 4-630CP-CTX (**57**) probes (Supplementary Figs. 21 and 22). Microtubules are well-defined tubular structures with a diameter of ~20 nm in mammalian cells. We exploited tubulin probes to evaluate the STED microscopy performance of the series of dyes (Supplementary Fig. 23). We found that the depletion efficiency (smaller $I_{sat}$) increases with the shift of the emission spectrum maximum towards the 775 nm STED laser (Fig. 5d, Supplementary Table 4). Such observations are in agreement with stimulated emission theory[41]. Fluorophores with smaller $I_{sat}$ can operate at lower STED laser powers, thus providing increased resolution and reducing potential phototoxicity. The smallest apparent FWHM of the microtubule equal to 21 ± 9 nm was obtained when living cells were stained with a 4-665SiR-CTX (**62**) probe (Fig. 5e, Supplementary Fig. 23 and Supplementary Table 4). This value is close to 16 ± 5 nm obtained using the MINFLUX method in fixed cells[42].

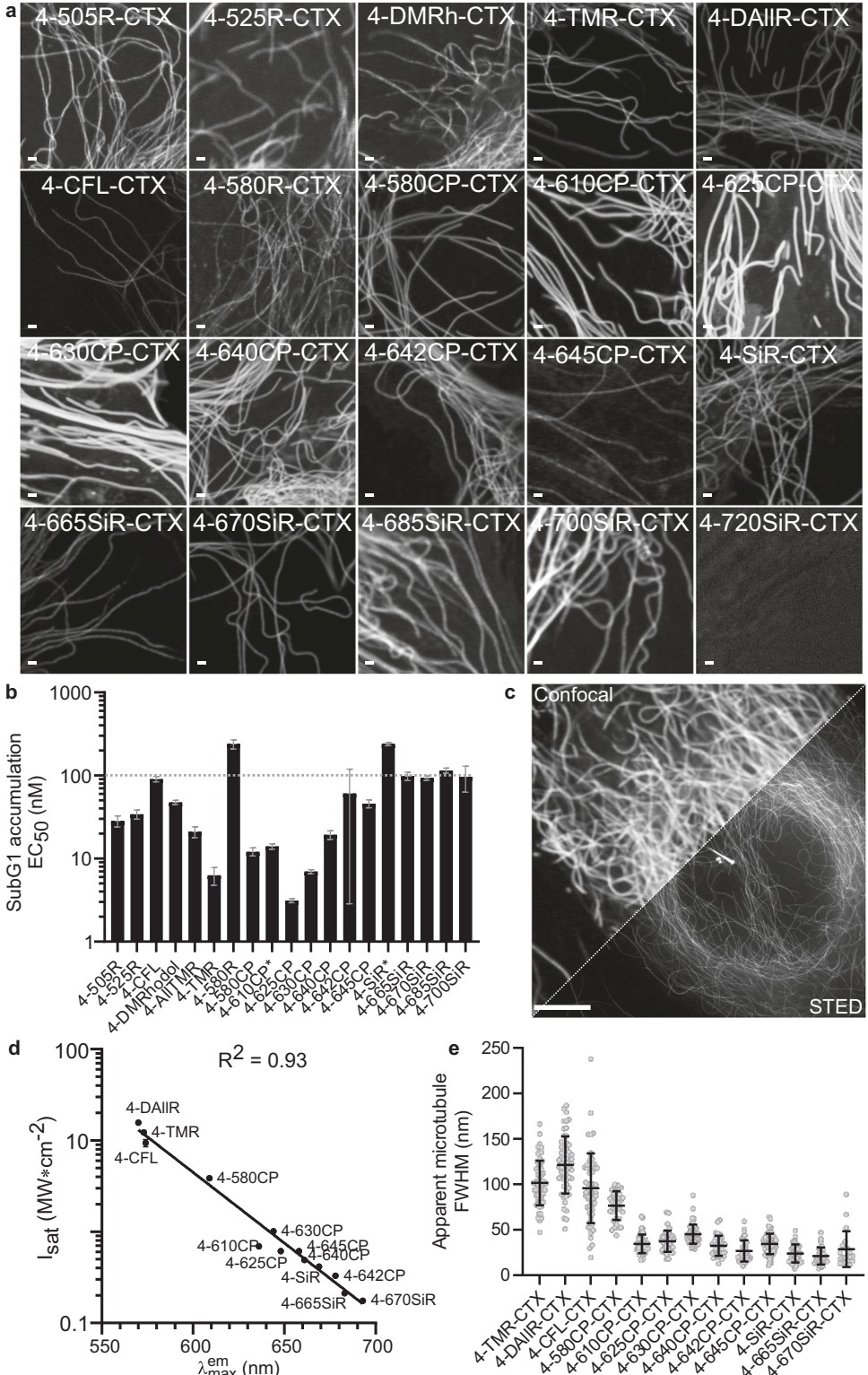

**Fig. 5 | Properties of fluorescent tubulin probes. a** Confocal images of microtubules in living human fibroblasts under no-wash conditions. Scale bar 1 μm. **b** Toxicity of tubulin probes for HeLa cells after 24 h of incubation. *EC₅₀ of 4-610CP-CTX (**55**) and 4-SiR-CTX (**61**) was measured in ref. [22]. Data represented as fitted half maximal effective concentration (EC₅₀) value with ± SEM, fitting of data points is shown in Supplementary Fig. 19c ($n = 3$ independent experiments). **c** Confocal and STED microscopy image of human fibroblasts stained with 10 nM 4-625CP-CTX (**56**) probe. Note, centrosome and primary cilium are visible in the center. Scale bar, 5 μm. **d** Plot showing the correlation of the stimulated emission efficiency ($I_{sat}$) and emission maximum ($\lambda^{em}_{max}$) of the fluorophore. $I_{sat}$ values represented fitted value with ± SEM, fitting is shown in Supplementary Fig. 23o ($n = 3$ independent experiments). Goodness of linear fitting is estimated by providing a coefficient of determination ($R^2$). **e** Measured of apparent microtubule full width at half maximum (FWHM) in STED mode with a 775 nm depletion laser set at 56 MW/cm², except 4-670SiR-CTX (**63**) was imaged using 28 MW/cm². Data represent the mean ± SD ($N = 30$). Source data are provided with this paper.

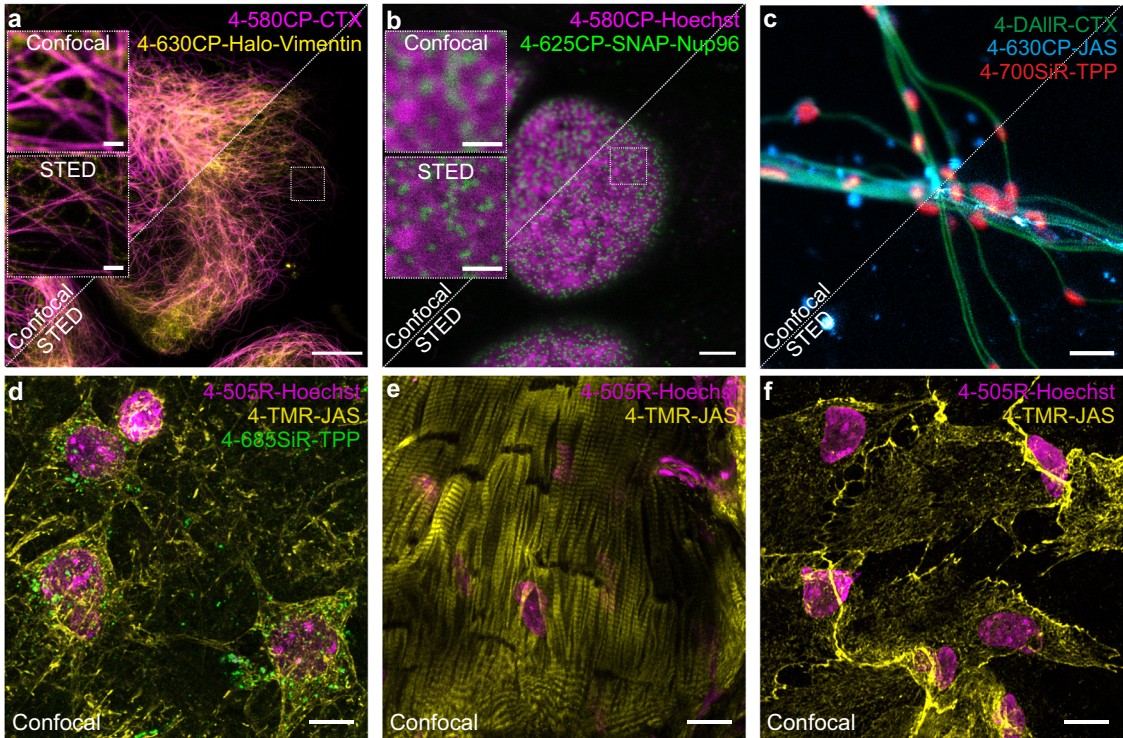

**Fig. 6 | Confocal and STED images of living cells and tissues stained with cell membrane-permeable probes. a** Confocal and STED images of U-2 OS cells expressing vimentin-Halo stained with 100 nM 4-630CP-(O$_4$)Halo (**70**) and 100 nM 4-580CP-CTX (**54**). Images acquired on an Abberior Expert Line microscope. Scale bars: large−10 μm, insets−1 μm. **b** Confocal and STED images of U-2 OS cells expressing Nup96-SNAP stained with 100 nM 4-625CP-BG (**107**) and 100 nM 4-580CP-Hoechst. Images acquired on an Abberior Expert Line microscope. Scale bars: large−3 μm, insets−1 μm. **c** Confocal and STED images of isolated mouse living neurons stained with 10 nM 4-DAllR-CTX (**52**), 10 nM 4-630CP-JAS (**103**), and 10 nM 4-700SiR-TPP (**86**) probes. Images were acquired on an Abberior Facility Line microscope. Scale bar, 1 μm. **d** Confocal images of brain living slices stained with 250 nM 4-505R-Hoechst (**88**) (nuclei-labeling probe, magenta), 1 μM 4-TMR-JAS (**101**) (actin-labeling probe, yellow), and 1.5 μM 4-685SiR-TPP (**85**) (mitochondria-labeling probe, green). **e** Heart and **f** liver living tissue slices stained with 250 nM 4-505R-Hoechst (**88**) (nuclei-labeling probe, magenta) and 1 μM 4-TMR-JAS (**101**) (actin-labeling probe, yellow). Images show max intensity projections of z-stacks. Scale bar: 10 μm. Images were deconvoluted with SVI Huygens software.

The slightly further redshifted 4-670SiR-CTX (**63**) probe already demonstrated a slightly worse resolution due to re-excitation by the 775 nm STED laser (Supplementary Fig. 23). The 3D STED imaging experiments demonstrated excellent performance of the 4-625CP-CTX (**56**) and 4-630CP-CTX (**57**) probes. The measured apparent FWHM was in the range of 100–150 nm along all axes (Supplementary Table 5 and Supplementary Fig. 20). Despite low fluorogenicity (based on a low $D_{50}$ value), most of the probes show a ratio above 10 of fluorescence signal increase on microtubules compared to the cytosol (Supplementary Table 4 and Supplementary Fig. 23). This observation highlights the highly specific staining of microtubules and high cell permeability of the probes.

### Other fluorescent probes
A wide spectral range of synthesized fluorophores opens up opportunities for multicolor imaging of many cell structures. We used the following targeting moieties: Hoechst for DNA, jasplakinolide (JAS) for actin, triphenylphosphonium (TPP) for mitochondria and pepstatin A (PepA) for lyso-somes (Fig. 3).

### Actin probes.
Our group and several other groups have reported fluorescent probes targeting actin, which are produced by con-necting jasplakinolide derivatives[43–45] via a long aliphatic linker to fluorescent dyes[46]. We hypothesized that the increased perme-ability of 4-isomers would eliminate the necessity for a long linker. Accordingly, we synthesized a series of probes by direct attach-ment of fluorescent dye to jasplakinolide derivatives (Fig. 3).

Indeed, no major differences in fluorescence signal intensity were observed in fibroblasts stained with a range of 4-610CP-JAS (**102**) and 4-610CP-C6-JAS concentrations (Supplementary Fig. 24). Inspired by this result, we synthesized actin probes spanning the full visible spectrum: 4-505R-JAS (**100**), 4-TMR-JAS (**101**), 4-630CP-JAS (**103**), and 4-685SiR-JAS (**104**). Confocal imaging of living fibroblasts confirmed good staining with all these probes (Sup-plementary Fig. 25a). A periodic actin ring pattern was clearly visible in the STED microscopy images of cultured mouse primary neurons stained with 4-610CP-JAS (**102**) or 4-630CP-JAS (**103**) (Fig. 6c and Supplementary Fig. 25b).

### DNA probes.
We used a previously reported design of DNA labeling probes using Hoechst linked via aliphatic C4-linker to the fluorescent dyes 4-505R-, 4-525R-, 4-CFL-, 4-625CP-, 4-630CP- and 4-685SiR- (**87**–**92**). All probes stained DNA in the nucleus of living fibroblasts except 4-685SiR-Hoechst (**92**), which showed high background in the cytosol (Supplementary Fig. 26).

### Mitochondria probes.
The matrix of the mitochondria has a pH ~ 8, which is higher than that of the cytosol (7.0–7.4), and the membrane potential that permits targeting by positively charged triphenylpho-sphonium (TPP)[47]. We synthesized probes by linking TPP via the C6-linker to the fluorescent dyes 4-525R-, 4-625CP-, 4-640CP-, 4-665SiR-, 4-685SiR- and 4-700SiR (**80**–**86**). All probes stained mitochondria except 4-525R-TPP (**80**), which demonstrated some off-targeting to intracellular vesicles and high background staining (Supplemen-tary Fig. 27).

**Lysosome probes**. In contrast to mitochondria, lysosomes are acidic organelles (pH ~ 5) harboring multiple proteases[48]. We synthesized lysosomal fluorescent probes by coupling fluorophores (4-525R-, 4-580R-, 4-630CP-, 4-642CP-, 4-SiR-, 4-685SiR- and 4-700SiR-) via an aliphatic linker to pepstatin A (**93**–**99**), a known inhibitor of cathepsin D protease. All studied probes featured selective lysosomal staining, and the most intense staining of lysosomal vesicles was obtained with 4-642CP-PepA (**96**), 4-SiR-PepA (**97**), and 4-685SiR-PepA (**98**) (Supplementary Fig. 28).

In summary, most of the newly generated probes targeting different cell structures and self-labeling tags efficiently stained living cells. The high success rate (>80%) confirms the advantage of incorporating isomer-4 rhodamines into fluorescent probe design.

## Multicolor imaging of living cells and tissues

Having in hand a set of fluorescent probes spanning the spectrum from 500 to 700 nm, we designed multicolor imaging experiments of living cells and tissue (Fig. 6). First, we combined 4-580CP-CTX (**54**) and 4-630CP-(O₄)-Halo (**70**) for two-color STED imaging of U-2 OS cells expressing vimentin Halo (Fig. 6a). Then, we used 4-625CP-BG (**107**) with 4-580CP-Hoechst[22] to stain U-2 OS cells expressing Nup96-SNAP. The obtained STED images clearly show that the Nup96-SNAP signal localizes to dark areas in the 4-580CP channel, confirming that the nuclear pore complexes protrude into the heterochromatin layer, creating chromatin exclusion zones (Fig. 6b).

Small molecule fluorescent probes are a particularly powerful tool for imaging primary cells and tissues[49]. To demonstrate this, we stained isolated mouse living hippocampal neurons using three probes targeting tubulin (4-DAllR-CTX (**52**)), actin (4-630CP-JAS (**103**)), and mitochondria (4-700SiR-TPP (**86**)). The three-color confocal and STED images clearly show mitochondria (red) embedded in the axon/dendrite cytoskeleton (Fig. 6c).

Another advantage of small molecule fluorescent probes is good penetration of the tissues. We demonstrated this by staining living isolated mouse brain, heart, and liver tissue sections with fluorescent probes targeting DNA, actin, and mitochondria (Fig. 6d–f).

Finally, the full palette of fluorescent dyes allowed us to design four color time-lapse live-cell experiments (Supplementary Movies 1–4 and Supplementary Table 6). To this end, we stained human fibroblasts and HUVECs with 4-505R-Hoechst (**88**), 4-DAllR-CTX (**52**), 4-642CP-PepA (**96**), and 4-700SiR-TPP (**86**). The obtained time-lapse movies show the dynamics of DNA, tubulin, lysosomes, and mitochondria with minimal bleaching (Supplementary Movies 1, 2). An alternative scheme employed 4-505R-CTX (**48**), 4-CFL-Hoechst (**47**), 4-624CP-PepA (**96**), and 4-700SiR-TPP (**86**) (Supplementary Movie 3). The third 4-color scheme consists of 4-505R-Hoechst (**88**), 4-DAllR-CTX (**52**), 4-625CP-(O₄)Halo (**69**), and 4-700SiR-TPP (**86**) and was devised to observe the dynamics of mitochondria in the network of vimentin and tubulin in U-2 OS cells expressing vimentin-Halo fusion (Supplementary Movie 4). In all cases, we took advantage of all four detectors on a commercial Abberior STED Facility line microscope and acquired data using a 2 + 2 simultaneous acquisition scheme. This allowed relatively fast imaging (3 s per frame) of large fields of view (50 × 50 μm) with minimal or no crosstalk between the channels.

## Discussion

We have devised a general, highly efficient, facile, and scalable strategy to synthesize 4-carboxyrhodamine dyes without the need for protecting groups. The aforementioned synthetic method allows gram-scale quantities of fluorescent dyes to be obtained from a single batch in a time-efficient and cost-effective manner. From the synthetic methodology perspective, we envision that this type of protecting group-free lithium–halogen exchange could be adapted to structurally similar compounds with protic functional groups at *ortho* positions that can form reasonably strong intramolecular hydrogen bond and significantly reduce the acidity of hydrogen-bonded protons. We used the whole set of obtained fluorescent dyes to generate a series of fluorescent probes targeting microtubules, actin, mitochondria, lysosomes, DNA, and Halo- and SNAP-tagged proteins. The isomer-4 rhodamine probes, due to the NGE phenomenon, demonstrate a combination of increased cell membrane permeability and specificity compared to commonly used isomers-5/6. These beneficial properties and resulting increased biocompatibility were retained with a large variety of fluorophores, as was demonstrated with a set of tubulin targeting probes. However, each probe should be viewed as a separate molecule with unique properties. It is not surprising that certain combinations of targeting moieties, linkers, and dyes, such as 4-685SiR-Hoechst (**92**) or 4-525R-TPP (**80**), might perform poorly due to the high complexity of the cellular environment, molecular crowding, off-targeting, metabolism or permeability issues. Furthermore, the relatively poor performance of 690SiR and 720SiR probes might be resulting from their extremely high hydrophobicity (spirolactone $c$Log$P \geq 8$). Despite this, such a high success rate in fluorescent probe development (>80%) is unique, as in general, it is very often that only a small subset of dye–ligand variations yields reasonably performing fluorescent probes for imaging living samples. The generality of such biocompatibility enhancement across a large structural variety of fluorescent probes allows fine-tuning to ideally meet the technical and biological requirements of confocal and STED imaging. A wide selection of probes operating at different wavelengths allows multicolor imaging of living specimens and tissues. We expect that the demonstrated high biocompatibility together with the scalable facile synthesis of rhodamine 4-isomers will serve as a general toolbox for scientists to improve the performance of existing fluorescent probes and will accelerate the creation of numerous new probes.

## Methods

### Ethical statement

All animal procedures were conducted in accordance with Directive 2010/63/EU of the European Parliament and the Council on the protection of animals used in research, as well as the Animal Welfare Law of the Federal Republic of Germany (Tierschutzgesetz der Bundesrepublik Deutschland, TierSchG). All mice were housed with a 12 h light/dark circadian cycle with ad libitum access to food and water.

### General synthesis procedure for compounds 2–26

In a 50 mL round-bottom flask, a degassed solution of 3-bromophthalic acid (100 mg, 0.408 mmol, 1 eq) in anhydrous THF (5 mL) was cooled to −78 °C (dry ice− acetone cooling bath). *n*-Butyllithium (640 μL, 1.6 M in hexane, 1.02 mmol, 2.5 eq) was introduced through a needle to the reaction mixture dropwise in 3–5 min. The color of the reaction mixture changed to light orange once−the last 0.5 eq was introduced. Once the addition was complete, stirring was continued at −78 °C for 20 min. Meanwhile, the corresponding ketones **K2**–**K21** (0.136 mmol, 0.33 eq) were dissolved in a minimal amount of THF (1–30 mL, varying depending on the ketone) and slowly injected into the reaction mixture with a syringe. The mixture was stirred for an additional 10 min at −78 °C, the cooling bath was removed, and the reaction mixture was allowed to warm to room temperature. Stirring was continued for 30 min at r.t., and then 1 mL of glacial acetic acid (1 mL, 17.5 mmol) was injected with a syringe, resulting in an immediate change in color. After stirring for 5 min, the THF was partially removed on a rotary evaporator, water (20 mL) and a solution of HCl were added (2 mL, 1 M), and the mixture was extracted with a DCM or DCM/MeOH (9:1) mixture (5 × 25 mL). The combined organic extracts were dried over $Na_2SO_4$ and filtered. The filtrate was concentrated on a rotary evaporator, the residue was deposited on celite, and the products were purified by flash column chromatography. See supplementary material for more details.

*NOTE: The reaction is highly sensitive to the ratio of the reagents used. Even a small amount of water can affect the concentration of alkyllithium reagents; therefore, we highly recommend using freshly distilled anhydrous THF and, if needed, predetermining the exact n-BuLi concentration prior to usage by titration.*

## Measurements of absorbance spectra in 1,4-dioxane–water mixtures

Measurements of the absorbance changes in 1,4-dioxane–water mixtures were performed by pipetting 2 μL of 1 mM stock solutions of dyes or probes in DMSO into a 96 bottom well plate (11 wells per sample) made from propylene (Corning 3364). To the wells going from right to left 300 μL of 1,4-dioxane–water mixtures containing 100%, 90%, 80%, 70%, 60%, 50%, 40%, 30%, 20%, 10%, or 0% of 1,4-dioxane (if needed, mixtures with 0.3% SDS are used, with an exception in 100% dioxane where SDS is not soluble). After incubation for 1 h at room temperature, the absorption of solutions in each well was recorded from 320 to 850 nm with a wavelength step size of 1 nm on a Spark® 20M (Tecan) multiwell plate reader. The background absorption of the propylene bottom plate was measured in wells containing only a 1,4-dioxane–water mixture with a similar amount of DMSO and subtracted from the spectra of the samples. $D_{50}$ values were obtained by fitting plots to the dose–response equation EC$_{50}$ (1) as implemented in GraphPad 9.2.0 software[50]

$$A = A_0 + (A_{max} - A_0)/\left(1 + \left(\frac{D_{50}}{d}\right)^{Hill}\right) \quad (1)$$

where $A_o$ is the absorbance at $\lambda_{max}$ at $\varepsilon_r = 0$ and $A_{max}$ is the highest absorbance at $\lambda_{max}$. $d$ is the dielectric constant of 1,4-dioxane–water mixture at a given point, Hill is the Hill slope coefficient determining the steepness of a dose–response curve, $D_{50}$ corresponds to $d$ value that provokes half of the absorbance amplitude ($A_{max} - A_0$).

## Determination of $K_{L–Z}$ values

Measurements of the absorbance changes in 1,4-dioxane–water 1:1 mixtures and ethanol + 0.1% TFA were performed by pipetting 2 μL of 1 mM stock solutions of dyes or probes in DMSO into a 96 propylene bottom well plate (Corning 3364) and adding 300 μL of the corresponding solvent. After incubation for 1 h at room temperature, the absorption of solutions in each well was recorded from 320 to 850 nm with a wavelength step size of 1 nm on a Spark® 20M (Tecan) multiwell plate reader. The background absorption of the propylene bottom plate was measured in wells containing only 1,4-dioxane–water and ethanol + 0.1% TFA mixtures with similar amounts of DMSO and subtracted from the spectra of the samples. The $K_{L–Z}$ was determined based on a procedure published before[19] and was calculated using the following equation: $K_{L–Z} = (\varepsilon_{dw}/\varepsilon_{max})/(1 - \varepsilon_{dw}/\varepsilon_{max})$, where $\varepsilon_{dw}$ is the extinction coefficient of the dyes/probes in a 1:1 dioxane:water solvent mixture. $\varepsilon_{max}$ refers to the maximal extinction coefficients measured in ethanol + 0.1% TFA. As noted previously[19,] the accurate determination of low $K_{L–Z}$ values ($\leq 10^{-3}$) is complicated by the relatively poor sensitivity of absorbance measurements. $K_{L–Z} = 10^{-3}$ was estimated if we observed a small but measurable absorbance signal in a 1:1 dioxane:water solvent mixture over the dye-free blank; $K_{L–Z}$ of $10^{-4}$ was estimated if we observed no measurable absorbance of the dye solution.

## Quantum chemical calculations

The initial geometries of the studied molecules were generated by using a molecular mechanics method (force field MMF94, steepest descent algorithm), and a systematic conformational analysis was carried out as implemented in Avogadro 1.1.1 software. The minimum energy conformer geometries found by molecular mechanics were further optimized with the Gaussian 09 program package[30] by means of density functional theory (DFT) using the Becke, 3-parameter, Lee–Yang–Parr (B3LYP)[51,52] exchange-correlation hybrid functional with the 6-311++G(d,p) basis set[53], including the polarizable continuum model[31] in water. Furthermore, harmonic vibrational frequencies were calculated to verify the stability of the optimized geometries. All the calculated vibrational frequencies were real (positive), indicating the true minimum of the calculated total potential energy of the optimized system. The computation was performed at the High-Performance Computing Center in Göttingen provided by GWDG.

The potential energy values for the $D_{50}$ simulation were obtained by performing geometry optimizations at the 6-311++G(d,p) IEFPCM (water) level of theory with additional input values of eps = 2.1, 5.7, 11.0, 18.2, 26.6, 35.2, 44.2, 53.3, 62.4, 71.4, and 80.4 to obtain 11 points against increasing dielectric constant. The calculations were performed for zwitterionic and spirolactone forms. Energy values were plotted against the dielectric constant, and the intersection point of the zwitterion and spirolactone plots was considered the simulated $D_{50}$ value. The calculations were performed for the free 4-TMR-COOH (**3**) dye and for the model 4-TMR-CONHMe compound with a truncated linker and ligand.

## Statistics and reproducibility

All microscopy imaging experiments were repeated two times on different non-consecutive days. Multiple fields of view ($n \geq 3$) are acquired during each imaging session and representative images are shown in the figures. All chemical synthesis procedures were repeated two times and both repetitions were successful.

## Reporting summary

Further information on research design is available in the Nature Portfolio Reporting Summary linked to this article.

## Data availability

The data supporting the findings of this study are available within the paper, Supplementary Information and Source Data file. Source data are provided with this paper.

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

## Acknowledgements

This study was funded by the Max Planck Society. The authors are grateful to Dr. Vladimir Belov, Jan Seikowski, Jens Schimpfhauser, and

Jürgen Bienert for NMR measurements of numerous samples and for providing the Boc-JAS actin targeting moiety. They also acknowledge Dr. Peter Lenart and Dr. Antonio Politi (Live-cell imaging facility) for the possibility of performing live-cell spinning disk confocal microscopy. We thank the head of the animal facility Dr. Ulrike Teichmann and the animal welfare officer Dr. Sarah Kimmina. J.B. is grateful to the Max Planck Society for a Nobel Laureate Fellowship. G.K. is grateful to EMBO for a Long-Term Fellowship (ALTF 135-2019). S.P. was supported by the Ph.D. program Genome Science—International Max Planck Research School. The authors acknowledge Jaydev Jethwa for his critical reading of the manuscript.

## Author contributions

J.B. and G.L. conceived and planned the study. J.B., G.K., R.G., T.K., K.A.K., S.P., and G.L. performed the experiments. J.B., K.A.K., S.P., G.K., R.G., and G.L. performed the data analysis. J.B. and G.L. wrote the initial draft; all authors contributed to the final version of the manuscript.

## Funding

## Competing interests

G.L. has filed a patent application (PCT/EP2011/064750, applicant EPFL, status: granted) on SiR derivatives. J.B., G.K., R.G., and G.L. have filed a patent (PCT/EP2021/053319, applicant Max Planck Society, status: pending) application on 4'-isomers of rhodamines. The remaining authors declare no competing interests.
