## [Peer Review File · Nature Communications]

A general highly efficient synthesis of biocompatible rhodamine dyes and probes for live-cell multicolor nanoscopyREVIEWER COMMENTS

Reviewer #1 (Remarks to the Author):

In this research, the author optimized the synthesis of 4-carboxyrhodamines reported in their previous work (Chem. Sci., 2020, 11, 7313–7323) and presented a facile protecting-group-free synthesis based on nucleophilic addition of lithium dicarboxybenzenide to the corresponding xanثone. The approach drastically reduced the synthesis difficulty, improved the overall yields and could be used to synthesize a wide range of symmetrical and unsymmetrical 4-carboxyrhodamines covering the whole visible spectrum. Applying these dyes, the author constructed a huge library of probes for targeted multiple structures in living cells, including microtubules, DNA, actin, mitochondria, lysosomes, Halo-tagged and SNAP-tagged proteins. These probes spanning the spectrum from 500 to 700 nm were finally used for multicolor confocal and STED imaging of living cells and tissues. Overall this work is interesting and data is supportive. Publication is recommended after the following concerns are addressed:

1. In Table 1, the conversion of 3-indophthalate suddenly improved so much when the amount of n-BuLi added was increased from 2.2 eq. to 2.5 eq.. Can the author give a reasonable explanation for this phenomenon? In addition, would the conversion of 3-indophthalate be higher if n-BuLi is added to 5 eq.?
2. The last column of second row in table 2 is 99:2. Is there some error? I am curious if the K2 conversion would be higher with 0.33 eq. or even less on the premise of 2 eq. or 2.5 eq. of n-BuLi. Similarly, under the premise of 0.33 eq. of K2, could the conversion of K2 be further improved when the addition of n-BuLi is increased to 3 eq.?
3. Please explain that almost all the probes based on 4-690SiR and 4-720SiR do not work.
4. In the manuscript, the author claimed that “The comparative row of staining intensities from the dimmest to the brightest is: 690SiR, 720SiR = no staining <<< 4-SiR < 665SiR ≤ 670SiR < 700SiR ≤ 685SiR < 642CP < 625CP ~ 630CP ~ 640CP-(O4)Halo (68-79)”. In the figure S18, just by observing at the naked eyes, I think the brightness of 4-665SiR-Halo was significantly stronger than 670SiR-Halo and 700SiR-Halo. So it is best for the author and reader to perform statistics on the intracellular brightness in figure S18 and give a quantitative result to confirm the statement.
5. The author states that “Our previous study showed that the incorporation of rhodamines’ 4-isomers enhances cell-membrane permeability of fluorescent probes. We were interested to see whether this conclusion is a general rule valid for a wide variety of fluorophores.”. However, the subsequent experiments did not compare the cell-membrane permeability of different types of rhodamines’ 4, 5 or 6-isomers. So please rephrase the sentence to make it more close to the experimental content.
6. Imaging data of 4-580R-CTX is lacking in Figure 5a.
7. In Figure S27, 4-525R-TPP showed high background staining. However, the author stated that it had some off-targeting to endosomes. Please verify the state though experiment or otherwise.

Reviewer #2 (Remarks to the Author):

Bucevičius et al. report synthesis of most popular class of rhodamine dyes using an exciting new chemical strategy. Their strategy allows addition of pendant phenyl rings to xanثones without protecting the (and later deprotecting) carboxylates. Typically, rhodamine syntheses requires protection of those carboxylic acids on the pendant phenyl ring. Further, they show that different conjugates can be added to the 4-carboxy using HATU coupling in one step. They authors prepared a library of such biologically relevant conjugates consisting of different colors and targeting different cellular organelles or regions. The imaging results are of good quality and I appreciate that authors provided tables with different parameters and ligand concentrations that they used for different imaging experiments.

I believe this manuscript should be strongly considered for publication at Nature communications as 4-carboxy rhodamine conjugates are novel scaffolds with limited synthesis literature and biological evaluation. However, before this article can be accepted, the authors must improve on the following in their manuscript. I have included the section headings provided by the authors to help them with my comments:

Minor comments:

1. Determination of Quantum Yields and Lifetimes

"The fluorescence quantum yields (absolute values) were obtained with a Quantaaurus-QY absolute PL quantum yield spectrometer (model C11347-12, Hamamatsu) according to the manufacturer's instructions".

Please provide more information on how the concentration (or approximate concentration) was chosen for determining the QY. What solvent/buffer and volume were used for determining the QY. What cuvettes were used for determining the QY? Similarly please provide more details for lifetime measurement experiments. Volume, concentration, solvent/buffer.

2. Plasmid Construction

"Construction and characterization of pEBTet GW_SNAP plasmid was characterized previously". Did the authors mean "Construction and characterization of pEBTet GW_SNAP plasmid was reported previously"?

3. Primary neuronal isolation and staining

"The protocol for hippocampal neurons isolation has been described previously in 3". "in" seems to be added by mistake.

4. Cell cycle analysis by imaging flow cytometry and EC50 determination

"loaded with ~30 μ l of the cell suspensions into a chamber of the slid.." I should be L. Please check for this in the rest of the manuscript as well.

5. Wide field microscope and imaging parameters

"For wide-filed fluorescence microscopy images were aquired..." It should be wide-field instead of wide-filed.

"Cy5 chennel". channel

What's the meaning for LED intensity numbers? What's the unit? Is that the number from the camera software?

6. Wide field microscope and imaging parameters

"Abberior STED Falicity Line scanning (Abberior Instruments GmbH)..". Facility?

7. General chemical experimental information and synthesis methods

"Agilent 400-MR spectrometer at 400.06 MHz (1H)" 1 in 1H should be in superscript.

9,9'-methylenebis(2,3,6,7-tetrahydro-1H,5H-pyrido[3,2,1-ij]quinolin-8-ol) (SI-1): "... give 0.7g (68%) of off-white solid. And was immediately used in the next step. The obtained compound S"

The last line should be corrected

For compound SI-2, What's the peak at 6.08 ppm in the 1H NMR? There is no need to re-take the NMR but if it is an impurity, it should be mentioned in the SI and labelled as such in the NMR spectrum.

6-Bromoindoline (SI-3): "...The reaction course was monitored by TLC hex:EtOAce (8:2)." Please fix this typo (EtOAc instead of EtOAce) at other places in the SI as well (e.g., SI-9).

The structure for compound 70 is missing on the NMR.

What are the peaks at 1.22 at 2.06 in the proton NMR for compound 14? They should be labelled as

impurity/grease.

8. In Table 2:1. Equivalents can be reported as 7.5 to 1 instead of 2.5 to 0.33.

Major comments:

1. For many molecules HRMS or even MS is not included. It is a standard practice in the field to provide HRMS of new molecules. Authors must report them.

2. NMRs for SI-16 and K-10 is not of sufficient quality. There are lots of peaks from impurities. I think either the authors should acquire these NMRs again or mention it explicitly in the synthesis protocol that impurities could not be separated. If the authors choose to report the impurities then yield should be changed to approximate yield. If these molecules were reported previously then provide references in the SI as well. This also holds for many other intermediates in the SI which have been reported by many labs (e.g., SI-24, 25 etc). The authors must explicitly mention it for each intermediate, if those have been reported previously.

3. ¹H for SI-17, SI-18, SI-24, K-6, K-8, K-9, K-18, K-21: The number of protons are mentioned as ratios. They should be corrected to account for exact number of protons. Similarly, include this on the NMR spectra to maintain consistency with the other spectra in the SI.

4. ¹H NMR for K6 should be reanalyzed to include the 6 protons from two methyl groups.

5. The authors should comment in the synthesis protocol about their reason to use pyridine-d₅ for NMR of compound 2. This is not a usual NMR solvent and the authors only used it in one case. The authors should consider acquiring this NMR in methanol or DMSO as they did for other fluorophores.

6. The authors have reported dioxane impurity in many of their proton NMRs. This is perplexing as the reaction or purification conditions did not use dioxane. How do authors know that the impurity is dioxane and not something else? What is the source of this dioxane? It's not the NMR solvents as it is not present in all of the NMRs. If it is not dioxane, then it is an impurity and it should be labelled as an unknown impurity. In theory, this will also reduce the reported yields as many NMRs have significant amount of this "dioxane" peak (e.g, compound 21, 23). Authors should provide integration values of these peaks in the reported spectra to allow quantification of this impurity by others. One way to make sure this is a solvent impurity is to include a LC/MS trace at 254 nm for compounds with the "dioxane" impurity.

7. The NMR for compound 7 has significant amount of solvent peaks. This should be reacquired after drying.

8. The authors should comment in the manuscript about their choice of using the peg4-HTL version instead of the standard peg2-HTL. The authors talk about the cellular permeability of conjugates at the 4-position but they don't include any kinetic evaluation. Since the authors have made the 4-conjugated version of SNAP and Halo, they should measure the rate of labeling with purified HaloTag or SNAP-tag. Typically, 6-conjugated Halo or SNAP versions are used as the Halo and SNAP-tag were evolved for dyes with 6-conjugated ligand. It would be important to know the kinetics of this labeling when using 4-conjugated version. I am not suggesting to measure it for all the conjugates but pick one from each type (O, CMe₂, SiMe₂) and compare it with the corresponding 6-analogs. Without these numbers, it is unbalanced to claim 4-conjugated versions are superior.

9. For all the fluorophore conjugated ligands, authors do not have sufficient characterizations. They have only provided ¹H NMR! I understand that ¹³C can be challenging when making small amounts. It is a standard practice to at least include a LC/MS trace at 254 to justify sufficient purity. Authors must include these for all the conjugated ligands.

10. This really bothered me: The yields for fluorophore conjugated ligands were determined spectroscopically. This is highly unusual. It is not incorrect but this can't be mentioned as yield in the typical sense. Authors must mention this explicitly in the main manuscript and the figure legend that the yields were determined in this fashion. Further, authors must report their protocol for obtaining the yields this way. How were the extinction coefficients of the "free-dye" measured? I ask this because the conjugated ligands were measured in 0.1% SDS.

11. To really claim that this is a general strategy for fluorophore synthesis, authors can consider synthesizing 6-analogs of rhodamines (O, CMe₂, SiMe₂). This would add tremendous value to the paper! Again, one example from each class would be enough.

9.

Reviewer #3 (Remarks to the Author):

Lukinavičius et al. reported a new synthesis strategy for rhodamines without the necessity of protecting groups. In this study, they optimized the lithium-halogen exchange reaction of 3-bromophthalic acid and efficiently obtained a library of oxygen, carbon, silicon bridged rhodamines and their labeling probes. Moreover, the imaging functionality of their probes was validated for STED imaging and multi-color time-lapse imaging. This work was solid with the successful development of a large quantity of effective probes. I admire the efforts the authors have paid on constructing such a library of rhodamines.

Concerns:

1. Is the strategy extendable to other bromo-substituted benzoic acid without neighboring group effect? The author might find it helpful to discuss the role of their neighboring group effect in the synthesis without protecting groups.
2. As far as I know, a major concern of this synthesis is the low yields of most xanthenes and silaxanthone in preparations, although the author might argue the high conversion ratio in the nucleophilic addition reaction and the structural diversity of amine groups in obtainable rhodamines.
3. In table 2, from entry 3 to entry 4, the conversion ratio is enhanced with the decrement of K₂, yet the selectivity is decreased. What could possibly be the reason behind this drop of selectivity?
4. For some probes, for instance, 4-690SiR-CP and 4-720SiR-CP, they demonstrate controversial results in bounding to SNAP-tag proteins in vivo and in vitro. In spectral studies, they demonstrated apparent fluorescence and absorption enhancements after incubating with SNAP-tag proteins (Supplementary Figure S11-12) whereas they failed to label SNAP proteins in SDS-PAGE analysis (Supplementary Figure S8) and in cellular studies (on Line 330). What might be the reason behind this inconsistency? Is it the aggregation of these probes that result in this failure? Is it possible to accommodate this failure with the substitution from CP to BG substrate? The same inconsistency was observed in 4-720SiR-(O₄)Halo and 4-690SiR-(O₄)Halo.
5. Some developed probes with neighboring group effect, for instance, 4-685SiR397-Hoechst and 4-525R-TPP failed to stain the designed targets. Please provide the discussion for these failures.
6. In Figure 6e and f, I could not find any existence of mitochondria in these images although the author stated that these issues were stained with "1.5 439 μM 4-685SiR-TPP (mitochondria-labelling probe, green)" (Line 439).
7. In table 1, n-BuLi was not properly written.
8. In Figure 4a, some of the probes D₅₀ value (4-SiR, 4-670SiR, etc.) could not be found. It seemed that they were covered by the D₅₀ value of corresponding dyes.
9. In Figure 5, the caption was mislabeled for d) and e).
10. In supplementary information of NMR spectra, the author should explain the inconsistency between the ¹H NMR spectra and the molecular structure of compound SI-2. The integral of proton peaks from ¹H NMR spectrum of SI-2 reveals 30 protons whereas the structure of SI-2 only has 28 protons. The author might find it helpful to check their NMR data.
11. Avoid using "excellent". Some typo errors (e.g., "2μLof " on Line 498).

Point-by-point response to the reviewers' comments:

Reviewer #1 (Remarks to the Author):

In this research, the author optimized the synthesis of 4-carboxyrhodamines reported in their previous work (Chem. Sci., 2020, 11, 7313–7323) and presented a facile protecting-group-free synthesis based on nucleophilic addition of lithium dicarboxybenzenide to the corresponding xanثone. The approach drastically reduced the synthesis difficulty, improved the overall yields and could be used to synthesize a wide range of symmetrical and unsymmetrical 4-carboxyrhodamines covering the whole visible spectrum. Applying these dyes, the author constructed a huge library of probes for targeted multiple structures in living cells, including microtubules, DNA, actin, mitochondria, lysosomes, Halo-tagged and SNAP-tagged proteins. These probes spanning the spectrum from 500 to 700 nm were finally used for multicolor confocal and STED imaging of living cells and tissues. Overall this work is interesting and data is supportive. Publication is recommended after the following concerns are addressed:

We are grateful to this reviewer for all comments/suggestions and the time spent reading our manuscript.

1. In Table 1, the conversion of 3-iodophthalate suddenly improved so much when the amount of *n*-BuLi added was increased from 2.2 eq. to 2.5 eq. Can the author give a reasonable explanation for this phenomenon? In addition, would the conversion of 3-iodophthalate be higher if *n*-BuLi is added to 5 eq.?

The reason why conversion of 3-iodophthalate suddenly improved could be related to high deviation at reaction conditions were 2.2 eq of *n*-BuLi was used. We have performed the reactions at quite small scale (50 mg of 3-bromophthalic acid) and even small error in amount of *n*-BuLi (we previously used 1.6 M) could have affected the result. Additionally, the reaction is highly sensitive towards dryness of the solvent, which could also have been a reason. Thus, we have performed the optimization step with 2.2 eq. of *n*-BuLi at larger scale (100 mg of 3-bromophthalic acid), with diluted 1.0 M *n*-BuLi and freshly dried/distilled THF. We have also performed the requested experiment with 5 eq. of *n*-BuLi and updated the table. We have updated Table 1:

Entry	Starting	n -BuLi eq.	A (%)*	B (%)*	C (%)*
1	I	1.0	69	trace	30
2	I	2.0	3	64	33
3	II	2.0	55	trace	44
4	II	2.2	20	25	55
5	II	2.5	4	55	41
6	II	3.0	1	57	42
7	II	5.0	0	64	36

*Conversion was determined by LC/MS analysis. Data in the table represents mean (N = 2).

2. The last column of second row in table 2 is 99:2. Is there some error? I am curious if the **K2** conversion would be higher with 0.33 eq. or even less on the premise of 2 eq. or 2.5 eq. of n-BuLi. Similarly, under the premise of 0.33 eq. of **K2**, could the conversion of **K2** be further improved when the addition of n-BuLi is increased to 3 eq.?

We thank reviewer for the insight. Indeed table 2 entry 2 contains an error it should be 99:1 instead of 99:2. We also agree with reviewer that more examples in optimization would give a reader better understanding of the presented synthesis method borderlines. As suggested, we additionally performed experiments with 2 eq. of n-BuLi and 0.33 eq. of **K2**, 3 eq. of n-BuLi and 0.33 eq. of **K2**, 2.5 eq. of n-BuLi and 0.167 eq. of **K2**. In addition, experiment with 2.5 eq. of n-BuLi and 0.5 eq. of **K2** was repeated to obtain a more precise result as this entry result was also stressed out by reviewer 3. We have updated the Table 2:

Entry	II, eq	K2, eq.	n-BuLi, eq.	Conversion of K2 (%)*	Ratio 2 : 1*
1	1	1	3	62	60:40
2	1	1	2	5	99:1
3	1	1	2.5	26	98:2
4	1	0.5	2.5	72	99:1
5	1	0.33	2.5	95	98:2
6	1	0.33	2	3	100:0
7	1	0.33	3	100	1:99
8	1	0.167	2.5	100	97:3

*Conversion and ratio was determined by LC/MS analysis. Data represents mean (n =2).

3. Please explain that almost all the probes based on **4-690SiR** and **4-720SiR** do not work.

We believe the reason might be that **4-690SiR** and **4-720SiR** dyes are very hydrophobic, show high D_{50} and low K_{L-Z} values (see Supplementary Table S2). This might result in the probes, which aggregate very efficiently in aqueous environment and the affinity to the target is not sufficient to disrupt formation of the aggregates. Furthermore, the structures of **4-690SiR** and **4-720SiR** contain 1,2-dihydroquinoline fragment which might result in offtargeting and poor performance in living cells. Definitive conclusion requires further dyes' library expansion coupled with detailed investigation which is beyond the scope of this manuscript and we are working on this currently.

4. In the manuscript, the author claimed that "The comparative row of staining intensities from the dimmest to the brightest is: 690SiR, 720SiR = no staining <<< 4-SiR < 665SiR ≤ 670SiR < 700SiR ≤ 685SiR < 642CP < 625CP ~ 630CP ~ 640CP-(O4)Halo (68-79)". In the figure S18, just by observing at the naked eyes, I think the brightness of 4-665SiR-Halo was significantly stronger than 670SiR-Halo and 700SiR-Halo. So it is best for the author and reader to perform statistics on the intracellular brightness in figure S18 and give a quantitative result to confirm the statement.

Indeed, we must agree with the reviewer, as our row of staining intensities is not quite right. Moreover, we realized that it might be a bit confusing for the reader: due to different spectra of the dyes, the fluorescence intensity upon excitation with a 640 nm laser does not reflect objectively on how good is a particular HaloTag substrate in terms of biocompatibility or labelling efficiency. To avoid such confusing comparison of the probes, we removed the comparative row from the main text. Instead, we suggested that a HaloTag substrate could be chosen based on photophysical characteristics. In our opinion, the purpose of figure S18 is to illustrate that most of the new 4-isomer based substrates efficiently stain vimentin-HaloTag in the cells rather than to compare the probes. Because of this, we also decided not to add quantification data in the figure S18.

5. The author states that “Our previous study showed that the incorporation of rhodamines’ 4-isomers enhances cell-membrane permeability of fluorescent probes. We were interested to see whether this conclusion is a general rule valid for a wide variety of fluorophores.”. However, the subsequent experiments did not compare the cell-membrane permeability of different types of rhodamines’ 4, 5 or 6-isomers. So please rephrase the sentence to make it more close to the experimental content.

We have rephrased a sentence in respect to reviewer comment, now it states: “We were interested to see whether biocompatibility and staining performance would be on the same level for a wider variety of 4-isomer based probes.”

6. Imaging data of **4-580R-CTX** is lacking in Figure 5a.

We have updated Figure 5a and Supplementary Figure S23 by including data of **4-580R-CTX** and **4-720SiR-CTX**.

7. In Figure S27, **4-525R-TPP** showed high background staining. However, the author stated that it had some off-targeting to endosomes. Please verify the state through experiment or otherwise.

We provided the zoomed-in insets in Figure S27, where off-targeting of the **4-525R-TPP** is clearly detectable. As we did not investigate the nature of the stained compartments in more detail, we modified the main text that now it reads: “...**4-525R-TPP**, which demonstrated some off-targeting to intracellular vesicles and high background staining”.

Reviewer #2 (Remarks to the Author):

Bucevičius et al. report synthesis of most popular class of rhodamine dyes using an exciting new chemical strategy. Their strategy allows addition of pendant phenyl rings to xanthenes without protecting the (and later deprotecting) carboxylates. Typically, rhodamine syntheses requires protection of those carboxylic acids on the pendant phenyl ring. Further, they show that different conjugates can be added to the 4-carboxy using HATU coupling in one step. The authors prepared a library of such biologically relevant conjugates consisting of different colors and targeting different cellular organelles or regions. The imaging results are of good quality and I appreciate that authors provided tables with different parameters and ligand concentrations that they used for different imaging experiments. I believe this manuscript should be strongly considered for publication at Nature communications as 4-carboxy rhodamine conjugates are novel scaffolds with limited synthesis literature and biological evaluation. However, before this article can be accepted, the authors must improve on the following in their manuscript. I have included the section headings provided by the authors to help them with my comments:

We are grateful to the reviewer for all comments/suggestions and the time spent reading our manuscript.

Minor comments:

1. Determination of Quantum Yields and Lifetimes “The fluorescence quantum yields (absolute values) were obtained with a Quantaaurus-QY absolute PL quantum yield spectrometer (model C11347-12, Hamamatsu) according to the manufacturer’s instructions”. Please provide more information on how the concentration (or approximate concentration) was chosen for determining the QY. What solvent/buffer and volume were used for determining the QY. What cuvettes were used for determining the QY? Similarly please provide more details for lifetime measurement experiments. Volume, concentration, solvent/buffer.

We have updated the descriptions of “Determination of absolute Quantum Yields” and the “Determination of fluorescence lifetimes” in the supporting information as follows:

Determination of absolute Quantum Yields:

All reported absolute fluorescence quantum yield values (Φ) were measured using a Quantaaurus-QY spectrometer (model C11374-01, Hamamatsu Photonics). This instrument uses an integrating sphere to determine photons absorbed and emitted by a sample. Measurements were carried out using dilute samples in air saturated solvents at 25°C at concentrations ranging from 10^{-6} to 10^{-7} M ($A < 0.1$ as indicated in the user’s manual) and by using 3 mL quartz cuvettes (Hamamatsu Photonics Art. No. A10095-02) provided by the instrument supplier. The fluorescence quantum yields were measured in PBS buffer (Table 3) and in PBS buffer containing 0.1% SDS (Table S1). Self-absorption corrections, if needed, were performed using the instrument software. Reported values are averages ($n = 3$) with standard deviation.

Determination of fluorescence lifetimes

The fluorescence decay characteristics of the solution samples in PBS (Table 3) or PBS + 0.1% SDS (Table S1) buffers at concentrations ranging from 10^{-6} to 10^{-7} M were recorded using a fluorescence lifetime

measurement system (Quantaaurus-Tau, Hamamatsu Photonics) in 3 mL high performance quartz glass cuvettes (Hellma Analytics Art. No. 101-10-K-40). The decay profile was registered for 53 - 100 ns interval after excitation and the experiment was continued until 10 000 peak count was reached. The instrument response function was obtained by using diluted LUDOX® TM-50 colloidal silica (Sigma Aldrich). The analysis of the obtained fluorescence decay profile was performed using the instrument software. Reported values are averages (n = 3) with standard deviation.

2. Plasmid Construction: "Construction and characterization of pEBTet GW_SNAP plasmid was characterized previously". Did the authors mean "Construction and characterization of pEBTet GW_SNAP plasmid was reported previously"?

Indeed, we meant that the construction and characterization of the plasmid was reported previously. We corrected this sentence as suggested.

3. Primary neuronal isolation and staining: "The protocol for hippocampal neurons isolation has been described previously in 3". "in" seems to be added by mistake.

We have corrected the sentence as suggested.

4. Cell cycle analysis by imaging flow cytometry and EC50 determination: "loaded with ~30 µl of the cell suspensions into a chamber of the slid.." I should be L. Please check for this in the rest of the manuscript as well.

We have corrected µl to µL and double-checked the text for similar mistakes.

5. Wide field microscope and imaging parameters "For wide-filed fluorescence microscopy images were acquired..." It should be wide-field instead of wide-filed.
"Cy5 chennel". Channel

We thank reviewer for the notice. We have corrected the sentences as suggested.

What's the meaning for LED intensity numbers? What's the unit? Is that the number from the camera software?

Indeed, that is the number from the camera software and the units are arbitrary units (A.U.). We have updated the table in supporting information accordingly.

6. Wide field microscope and imaging parameters "Abberior STED Falicilty Line scanning (Abberior Instruments GmbH).. " Facility?

The name of the microscope was corrected.

7. General chemical experimental information and synthesis methods "Agilent 400-MR spectrometer at 400.06 MHz (1H)" 1 in 1H should be in superscript.

Corrected as indicated.

8. 9,9'-methylenebis(2,3,6,7-tetrahydro-1H,5H-pyrido[3,2,1-ij]quinolin-8-ol) (SI-1): "... give 0.7g (68%) of off-white solid. And was immediately used in the next step. The obtained compound S"
The last line should be corrected

The last line was corrected accordingly.

9. For compound **SI-2**, What's the peak at 6.08 ppm in the 1H NMR? There is no need to re-take the NMR but if it is an impurity, it should be mentioned in the SI and labelled as such in the NMR spectrum.

We thank reviewer for raising this point. We found that the compound **SI-2** is rather unstable in CDCl₃ resulting in appearance of this broad peak at approximately 6.08 ppm, but we did not investigate it further. As a response to the raised point, we provide the NMR data of **SI-2** obtained in CD₂Cl₂. We have updated the supporting information and added a copy of the NMR data.

10. 6-Bromoindoline (**SI-3**): "...The reaction course was monitored by TLC hex:EtOAc (8:2)." Please fix this typo (EtOAc instead of EtOAc) at other places in the SI as well (e.g., **SI-9**).

We have corrected all typos related to the mentioned case throughout the text.

11. The structure for compound 70 is missing on the NMR.

Indeed, we have added a structure of compound **70** on the NMR.

12. What are the peaks at 1.22 at 2.06 in the proton NMR for compound **14**? They should be labelled as impurity/grease.

Labels were added to proton NMR for compound **14** accordingly: 1.22 ppm - grease, 2.06 ppm – acetonitrile.

13. In Table 2: 1. Equivalentents can be reported as 7.5 to 1 instead of 2.5 to 0.33.

We agree that for the sake of clarity were has to be a reagent reported as 1 equivalent. In our case it was the reagent **II**, which was not indicated in the table 2. We inserted a column for reagent **II** and clearly indicate that the amount of reagent **II** is 1 equivalent. The table 2 was updated accordingly.

Major comments:

1. For many molecules HRMS or even MS is not included. It is a standard practice in the field to provide HRMS of new molecules. Authors must report them.

We added HRMS/ESI data for all compounds described in supporting information (previously known and new).

2. NMRs for **SI-16** and **K-10** is not of sufficient quality. There are lots of peaks from impurities. I think either the authors should acquire these NMRs again or mention it explicitly in the synthesis protocol that impurities could not be separated. If the authors choose to report the impurities then yield should be changed to approximate yield. If these molecules were reported previously then provide references in the SI as well. This also holds for many other intermediates in the SI which have been reported by many labs (e.g., SI-24, 25 etc). The authors must explicitly mention it for each intermediate, if those have been reported previously.

Indeed, the quality of NMRs of compounds **SI-16** and **K10** were not the best quality. We have updated the supporting information with the better quality NMR spectra for compounds **SI-16** and **K10**. We have repeated the synthesis of compound **SI-16** and updated the yield. We fully agree with the reviewer that it is crucial to provide a reference of the previously reported compounds. We address this by referencing all previously reported intermediate compounds and it is stated that the compound was reported previously. We chose to report synthesis of some known intermediates for the sake of better reproducibility and better clarity for readers and we provided synthetic procedures starting from commercially available precursors rather than from the first unreported compound.

3. ^1H for **SI-17**, **SI-18**, **SI-24**, **K-6**, **K-8**, **K-9**, **K-18**, **K-21**: The number of protons are mentioned as ratios. They should be corrected to account for exact number of protons. Similarly, include this on the NMR spectra to maintain consistency with the other spectra in the SI.

We have performed the requested corrections.

4. ^1H NMR for **K6** should be reanalyzed to include the 6 protons from two methyl groups.

Indeed, we have missed to include protons from the two methyl groups. We have corrected the NMR characterization and provided new copy of ^1H NMR. We thank reviewer for the remark.

5. The authors should comment in the synthesis protocol about their reason to use pyridine- d_5 for NMR of compound **2**. This is not a usual NMR solvent and the authors only used it in one case. The authors should consider acquiring this NMR in methanol or DMSO as they did for other fluorophores.

This was a special case we agree that d_5 -pyridine is not standard and costly solvent. We used d_5 -pyridine in order to obtain NMR in a spirolactone state (closed state) of compound **2**. In d_6 -DMSO the ^1H proton

peaks were very broad and unresolved, ^{13}C resolution was extremely low. The likely reason for this is a dynamics equilibrium between spirolactone and zwitterion at the NMR time scale. Alternatively, CD_3OD containing CF_3COOD additive pushes the equilibrium towards zwitterion which helps to get a good ^1H NMR resolution, but we could not obtain a reasonable quality ^{13}C NMR. To resolve this issue, now we have registered ^{13}C NMR at higher concentrations and with higher number of scans on 400MHz NMR system equipped with a cryoprobe, which enhances a signal up to 5-fold. We updated the supporting information and now we provide both NMRs (in d_5 -pyridine and in CD_3OD).

6. The authors have reported dioxane impurity in many of their proton NMRs. This is perplexing as the reaction or purification conditions did not use dioxane. How do authors know that the impurity is dioxane and not something else? What is the source of this dioxane? It's not the NMR solvents as it is not present in all of the NMRs. If it is not dioxane, then it is an impurity and it should be labelled as an unknown impurity. In theory, this will also reduce the reported yields as many NMRs have significant amount of this "dioxane" peak (e.g, compound **21**, **23**). Authors should provide integration values of these peaks in the reported spectra to allow quantification of this impurity by others. One way to make sure this is a solvent impurity is to include a LC/MS trace at 254 nm for compounds with the "dioxane" impurity.

We agree that the presence of 1,4-dioxane in current context is perplexing. As for a standard protocol, obtained dyes were lyophilized from MeCN/ H_2O mixture, but in some special cases, this type of lyophilisation failed to provide a dye in a powder form, instead yielding some of the dyes in a consistency of thick oil. In these cases, we performed a second lyophilisation from 1,4-dioxane to obtain dyes in form of powder. We have used a secondary lyophilisation from 1,4-dioxane before determining yields for the following compounds: **4-685SiR-COOH**, **4-720SiR-COOH**, **4-525R-COOH**, **4-580CP-COOH** and compound **21**. It is reported previously that after lyophilisation some rhodamine dyes are obtained in a complex form with dioxane. To address this point, we have updated experimental procedures and now we indicate the use of 1,4-dioxane. We have quantified the present amount of 1,4-dioxane and we report complex ratio in the experimental procedures. The determined rhodamine-dye:1,4-dioxane complex ratios are determined as follows: 6:1 – **4-720SiR-COOH**; 12:1 - **4-685SiR-COOH**; 1:1 - compound **21**; 13:1 - **4-525R-COOH**; 5:1 - **4-580CP-COOH**. We have recalculated the yields for aforementioned dyes and updated values in the main text, figure 2 and supporting information accordingly. The recalculated yields now are as follows: for **4-720SiR-COOH** yield changed from 67% to 65%; for **4-685SiR-COOH** from 55% to 54%; for compound **21** from 91% to 75%; for **4-525R-COOH** from 72% to 71% and for **4-580CP-COOH** from 89% to 85%.

7. The NMR for compound **7** has significant amount of solvent peaks. This should be reacquired after drying.

The high amount of DMSO was left in a sample after the solvent switch from d_6 -DMSO (washing with DMSO) to CD_3OD . As suggested, we reacquired NMR and provide a better quality spectra for compound **7**.

8. The authors should comment in the manuscript about their choice of using the peg4-HTL version instead of the standard peg2-HTL. The authors talk about the cellular permeability of conjugates at the 4-position but they don't include any kinetic evaluation. Since the authors have made the 4-conjugated version of SNAP and Halo, they should measure the rate of labeling with purified HaloTag or SNAP-tag. Typically, 6-conjugated Halo or SNAP versions are used as the Halo and SNAP-tag were evolved for dyes with 6-conjugated ligand. It would be important to know the kinetics of this labeling when using 4-conjugated version. I am not suggesting to measure it for all the conjugates but pick one from each type (O, CMe₂, SiMe₂) and compare it with the corresponding 6-analogs. Without these numbers, it is unbalanced to claim 4-conjugated versions are superior.

We agree with the reviewer that kinetic evaluation of SNAP/Halo-tag substrates is interesting aspect of SNAP/Halo-tag protein characterization. However, our study is mainly focusing on the properties of fluorescent dyes and their biocompatibility. Indeed, we are intending to solve crystal structures of 4-isomers in complex SNAP/Halo-tag proteins and provide rhodamine structure to function relation. However, we feel that it's a time-consuming separate study. Thus, we have toned down our statements about superiority of 4-isomers' substrates for SNAP/Halo-tag. In principle, what we wanted to highlight is that 4-isomer probes are also compatible with aforementioned tags.

9. For all the fluorophore conjugated ligands, authors do not have sufficient characterizations. They have only provided ¹H NMR! I understand that ¹³C can be challenging when making small amounts. It is a standard practice to at least include a LC/MS trace at 254 to justify sufficient purity. Authors must include these for all the conjugated ligands.

Indeed, the amounts of the fluorescent probes obtained was not sufficient to register reasonable quality of ¹³C NMR spectra. However, we fully agree that additional sample purity justification might be important and could contribute significantly for reproducibility of the results. To address this issue we have provided LC/MS trace at 254 nm (in addition, if possible also trace at dye absorbance maximum), mass and UV spectra in the supporting information for all of the 83 reported fluorescent probes.

10. This really bothered me: The yields for fluorophore conjugated ligands were determined spectroscopically. This is highly unusual. It is not incorrect but this can't be mentioned as yield in the typical sense. Authors must mention this explicitly in the main manuscript and the figure legend that the yields were determined in this fashion. Further, authors must report their protocol for obtaining the yields this way. How were the extinction coefficients of the "free-dye" measured? I ask this because the conjugated ligands were measure in 0.1% SDS.

We understand the misapprehension of the method we used, but we must say that yield determination based on absorption is common practice once working with small amounts of strongly absorbing compounds. To address this issue, we have provided a detailed description of how we have measured yields in a spectroscopic manner. We feel obliged to emphasize that we found the spectroscopic yield determination method to be more precise than the conventional weighting approach, especially if sub-milligram amount of fluorescent probe is obtained. For this reason, we always characterize new dyes in PBS containing 0.1% SDS (Supplementary Table S1) so that we could use the obtained extinction

coefficient for the reaction yield calculation. Some fluorophore conjugates trends to aggregate in PBS buffer and addition of 0.1% of SDS detergent (sodium dodecyl sulfate) dissolves the aggregates. SDS additive might slightly affect the extinction coefficient (in respect to extinction measured in PBS) and we report all extinction coefficients and other photophysical properties determined in PBS (Table 3) and in PBS + 0.1% SDS additive (Supplementary Table S1). We have found that for some fluorescent probes, based on **4-CFL**, **4-DMRh**, **4-SiR**, **4-670SiR**, **4-690SiR** and **4-720SiR**, the extinction coefficient of the free dye and a conjugate is not similar due to the higher propensity of the conjugates of these dyes to exist in spiro lactone form even in the presence of SDS additive. Thus, for the conjugates of aforementioned dyes, the reaction yields were determined by a conventional weighting approach. We clearly state in the experimental section of the supporting information, if the yield of probe was determined spectroscopically or by conventional weighting approach. In addition, we have updated the supporting information with a description of “Reaction yield determination by absorption spectroscopy” method, see below:

Reaction yield determination by absorption spectroscopy:

After the purification and removal of the solvent, the obtained fluorescent probes were dissolved in a precise (700 μL) volume of the d_6 -DMSO solvent and were transferred to NMR tube to obtain ^1H spectra. Afterwards the contents of the NMR tubes were transferred to an eppendorf and was considered as stock solution. Five 2 μL samples were taken from stock solution and were diluted in five separate eppendorf's with 98 μL of PBS + 0.1% SDS (50-fold dilution), vigorously mixed and aged for 30 min to dissolve aggregates. Then absorption of 2 μL of the diluted samples were measured on nanodrop (Nanodrop 1000, Peqlab) with 1 mm optical path. The measured absorption intensity values at the dyes absorption maxima value were averaged and concentration of the stock solution was determined according to the equation:

$$C = \frac{\text{dilution} * A}{\epsilon * l}$$

where C – concentration of stock solution; A – sample absorption, ϵ - extinction coefficient of the dye in PBS containing 0.1% SDS, l – path length

Once the concentration of the stock solution is measured the mass of the obtained fluorescent conjugate could be calculated by following equation:

$$m = C * MW * V$$

where C – concentration of stock solution; MW – molecular weight of the compound; V- volume of stock solution.

Finally, the yield of the synthesis step could be determined by the classical equation:

$$\text{Yield \%} = \frac{m}{m_{\text{teor}}} * 100$$

Where m – obtained mass of the isolated product; m_{teor} – maximal theoretical mass of the product in the reaction.

11. To really claim that this a general strategy for fluorophore synthesis, authors can consider synthesizing 6-analogs of rhodamines (O, CMe₂, SiMe₂). This would add tremendous value to the paper! Again, one example from each class would be enough.

We fully agree with a reviewers comment and we tried similar conditions with 2-bromo-terephthalic acid, which would yield 6-carboxy analogs, and with 4-bromoisophthalic acid, which would yield 5-carboxy analogs. However, the reaction is not yielding desired products because the acidity of the second carboxy group is not reduced enough in these compounds as compared to 3-bromophthalic acid. Once n-BuLi is added to the solution of 2-bromo-terephthalic acid or 4-bromoisophthalic acid, the formed lithium carboxylate salts immediately crash out of solution and prevents formation of lithium benzenide species. Our developed synthesis protocol is optimized and oriented to the protecting group free synthesis of 4-carboxy rhodamines including the rhodamine dyes, which were not accessible, by other approaches. From the synthetic methodology perspective, we envision that this type of protecting group free lithium-halogen exchange probably could be adapted to structurally similar compounds with protic functional groups at *ortho* positions that can form reasonably strong intramolecular hydrogen bond. Additionally, we updated the main text with short discussion related to the importance of neighboring group effect on the protecting group free lithiation reaction.

Reviewer #3 (Remarks to the Author):

Lukinavičius et.al. reported a new synthesis strategy for rhodamines without the necessity of protecting groups. In this study, they optimized the lithium-halogen exchange reaction of 3-bromophthalic acid and efficiently obtained a library of oxygen, carbon, silicon bridged rhodamines and their labeling probes. Moreover, the imaging functionality of their probes was validated for STED imaging and multi-color time-lapse imaging. This work was solid with the successful development of a large quantity of effective probes. I admire the efforts the authors have paid on constructing such library of rhodamines.

We are grateful to this reviewer for all comments/suggestions and the time spent reading our manuscript.

Concerns:

1. Is the strategy extendable to other bromo-substituted benzoic acid without neighboring group effect? The author might find it helpful to discuss the role of their neighboring group effect in the synthesis without protecting groups.

Indeed, we tried similar conditions with 2-bromo-terephthalic acid, which would yield 6-carboxy analogs, and with 4-bromoisophthalic acid, which would yield 5-carboxy analogs, however the chemistry does not work with these compounds. The reason is that there is no reduction of acidity of the second carboxy group in these compounds as in 3-bromophthalic acid. Therefore, once n-BuLi is added to the solution of 2-bromo-terephthalic acid or 4-bromoisophthalic acid the formed lithium carboxylate salts immediately crash out of solution preventing formation of lithium benzenide species. We thank reviewer for the suggestion and we have updated the main text with short discussion on importance of neighboring group effect on the protecting group free lithiation reaction perspective.

2. As far as I know, a major concern of this synthesis is the low yields of most xanthenes and silaxanthone in preparations, although the author might argue the high conversion ratio in the nucleophilic addition reaction and the structural diversity of amine groups in obtainable rhodamines.

The synthesis of xanthenes, carboxanthenes or silaxanthenes is highly diverse depending on their molecular structure. Each xanthone synthesis has to be started from different commercially available starting materials resulting in variable number of the synthesis steps and the yields. We have not optimized synthesis of xanthenes in this manuscript, but focused on developing protecting group free 4-carboxy rhodamines synthesis protocol. We would like to emphasize that high conversion ratio in the nucleophilic addition reaction also reduces the need of higher quantities of xanthenes.

3. In table 2, from entry 3 to entry 4, the conversion ratio is enhanced with the decrement of K2, yet the selectivity is decreased. What could possibly be the reason behind this drop of selectivity?

We think that the reason is high reaction sensitivity to any excess on n-BuLi and we performed optimisations at rather small scale (starting from 50 mg of 3-Bromophthalic acid) and using 1.6 M n-BuLi, and one of the attempts might have had a higher deviation to n-BuLi excess thus resulting in loss of selectivity. We have repeated the optimisations at higher reaction scale and with more dilute n-BuLi (1 M). We have updated the table 2:

Entry	II, eq	K2, eq.	n-BuLi, eq.	Conversion of K2 (%)*	Ratio 2 : 1*
1	1	1	3	62	60:40
2	1	1	2	5	99:1
3	1	1	2.5	26	98:2
4	1	0.5	2.5	72	99:1
5	1	0.33	2.5	95	98:2
6	1	0.33	2	3	100:0
7	1	0.33	3	100	1:99
8	1	0.167	2.5	100	97:3

*Conversion and ratio was determined by LC/MS analysis. Data represents mean (n=2).

4. For some probes, for instance, **4-690SiR-CP** and **4-720SiR-CP**, they demonstrate controversial results in bounding to SNAP-tag proteins in vivo and in vitro. In spectral studies, they demonstrated apparent fluorescence and absorption enhancements after incubating with SNAP-tag proteins (Supplementary Figure S11-12) whereas they failed to label SNAP proteins in SDS-PAGE analysis (Supplementary Figure S8) and in cellular studies (on Line 330). What might be the reason behind this inconsistency? Is it the aggregation of these probes result in this failure? Is it possibly to accommodate this failure with the substitution from CP to BG substrate? The same inconsistency was observed in **4-720SiR-(O4)Halo** and **4-690SiR-(O4)Halo**.

The increase of absorption is very low after addition of SNAP-Tag protein (red line) to **4-690SiR-CP** and **4-720SiR-CP** compounds shown in supplementary Figure S11. The fluorescence measurements, shown in Figure S12, are a lot more sensitive and the increase in signal better defined, but the signal intensity is still very low. In contrast, the absorption or fluorescence signals are much more intensive if 0.1% SDS is added to the substrate solution. This indicates that the labeling of SNAP-tag protein is extremely low and corresponding fluorescent bands cannot be seen or are very weak in SDS-PAGE analysis. Indeed, **4-690SiR** and **4-720SiR** dyes have very high D_{50} and low K_{L-2} values meaning that they tend to exist in non-fluorescent spiro lactone form which tends to aggregate. The poor labeling is likely to be attributed to a strong aggregation of the probes in PBS and even addition of SNAP-tag does not disrupt these aggregates. In respect to **4-720SiR-(O4)Halo** and **4-690SiR-(O4)Halo** probes, the increase fluorescence after addition of Halo-tag protein in vitro is significantly higher compared to increase for -CP substrates, thus the labeling becomes visible in SDS-PAGE analysis at least for **4-690SiR-(O4)Halo** substrate. However, the labeling was still too poor in cellular environment resulting in poor quality microscopy images. As most

of the probes derived from **4-690SiR** and **4-720SiR** performed poorly in terms of image quality we believe that switching from CP to BG substrate might not be enough.

5. Some developed probes with neighboring group effects, for instance, **4-685SiR-Hoechst** and **4-525R-TPP** failed to stain the designed targets. Please provide the discussion for these failures.

We agree with the reviewer that the poor performance of the probes should be explained. It should be mentioned that each probe should be viewed as a separate molecule with unique properties. It is not surprising that certain combinations of targeting moiety, linker, and dye might perform poorly due to the high complexity of the cellular environment, molecular crowding, off-targeting, metabolism, or permeability issues. We are attempting to address these issues by sampling a large variety of fluorescent dyes, but such a study is beyond the scope of this manuscript. We have updated the manuscript discussion to highlight this aspect of the probe design.

6. In Figure 6e and f, I could not find any evidence of mitochondria in these images, although the author stated that these tissues were stained with "1.5 μ M **4-685SiR-TPP** (mitochondria-labelling probe, green)" (Line 439).

As correctly pointed out, we updated the figure legend for clarity as follows:

"Confocal images of d) brain samples stained with 250 nM **4-505R-Hoechst** (nuclei-labelling probe, magenta) and 1 μ M **4-TMR-JAS** (actin-labelling probe, yellow) and 1.5 μ M **4-685SiR-TPP** (mitochondria-labelling probe, green). e) Heart and f) liver tissue sections stained with 250 nM **4-505R-Hoechst** (nuclei-labelling probe, magenta) and 1 μ M **4-TMR-JAS** (actin-labelling probe, yellow). Images show maximum intensity projections of z-stacks. Scale bar: 10 μ m. Images were deconvoluted with SVI Huygens software."

7. In table 1, n-BuLi was not properly written.

We have corrected this mistake.

8. In Figure 4a, some of the probes' D₅₀ values (**4-SiR**, **4-670SiR**, etc.) could not be found. It seemed that they were covered by the D₅₀ value of corresponding dyes.

Indeed, the values were covered as the D₅₀ values of these dyes and probes are higher than 80, which is above the scale of the evaluation by this approach as it is limited by the dielectric constant of water. We updated Figure 4a with some additional color indication and we state in the text that numerical values can be found in supplementary table 2.

9. In Figure 5, the caption was mislabeled for d) and e).

We have corrected the captions.

10. In supplementary information of NMR spectra, the author should explain the inconsistency between the ¹H NMR spectra and the molecular structure of compound SI-2. The integral of proton peaks from ¹H NMR spectrum of SI-2 reveals 30 protons whereas the structure of SI-2 only has 28 protons. The author might find it helpful to check their NMR data.

Indeed, it was our mistake and the additional two protons should not have been integrated for the report summary. We have noticed that this compound is not very stable in CDCl₃ giving rise of the broad peak at approximately 6.08 ppm, but we did not investigate it further. Thus, we provided new better quality NMR data of **SI-2** recorded in CD₂Cl₂. We have updated the supporting information and added a copy of the NMR data.

11. Avoid using "excellent". Some typo errors (e.g., "2μLof " on Line 498).

We have minimized the use of "excellent" in the text and we have corrected the aforementioned typo error.

REVIEWERS' COMMENTS

Reviewer #1 (Remarks to the Author):

In the revised manuscript, the author makes the detailed explanations and supplementary experiments for responding to the questions raised. However, there are still some issues to be further clarification:

1: In the main text, the author claimed that spectral tuning reduces the need for high STED laser power. Why can the spectral tuning reduce the need for high STED laser power? A clearer statement and the corresponding references are needed.

2: The author argues that almost all the probes based on 4-690SiR and 4-720SiR do not work is due to the strong hydrophobic of 4-690SiR and 4-720SiR. However, D50 and KL-Z can only reflect that dyes tend to be in the fluorescent zwitterion or non-fluorescent spirolactone, and cannot illustrate the hydrophobicity of dyes. LogP value of dyes need to be added to support the statement. In addition, 4-690SiR and 4-720SiR could effectively aggregate in aqueous environment. the author needs to add particle size analysis or other experiments for supporting the statement.

Reviewer #2 (Remarks to the Author):

I am satisfied with the additions and corrections that the authors have provided. I want to thank the authors for the effort they put in making these changes. I recommend it for publication without any additional comments.

Reviewer #3 (Remarks to the Author):

The authors have solved all my concerns. In this work, they have successfully developed a synthetic strategy to improve the overall yields of 4-carboxylrhodamines, and followingly gained a library of new fluorophores covering the full color palette. They have wisely constructed the linkers for their fluorophores to obtain a huge number of probes for multi-color imaging proteins and sub-cellular organelles. Overall, I am convinced by their efforts and recommend the publication of this study in Nature Communications.

Point-by-point response to the reviewers' comments:

Reviewer #1 (Remarks to the Author):

In the revised manuscript, the author makes the detailed explanations and supplementary experiments for responding to the questions raised. However, there are still some issues to be further clarification:

We are particularly grateful to this reviewer for the comments/suggestions and the time spent reading our manuscript.

1: In the main text, the author claimed that spectral tuning reduces the need for high STED laser power. Why can the spectral tuning reduce the need for high STED laser power? A clearer statement and the corresponding references are needed.

We thank for the remark and we would like to explain our statement. Spectral tuning allows to change the distance between depletion laser wavelength (commonly used 775 nm) and fluorescent dye emission maximum. This results in less depletion laser energy needed to achieve the same extent of stimulated emission. We demonstrate this phenomenon experimentally in Figure 5d. To further clarify our statement, we have modified phrasing in the main text accordingly.

2: The author argues that almost all the probes based on 4-690SiR and 4-720SiR do not work is due to the strong hydrophobic of 4-690SiR and 4-720SiR. However, D50 and KL-Z can only reflect that dyes tend to be in the fluorescent zwitterion or non-fluorescent spiro lactone, and cannot illustrate the hydrophobicity of dyes. LogP value of dyes need to be added to support the statement. In addition, 4-690SiR and 4-720SiR could effectively aggregate in aqueous environment. the author needs to add particle size analysis or other experiments for supporting the statement.

We are grateful for the suggestions and fully agree that D_{50} and K_{L-Z} are not reflecting the hydrophobicity of dyes. We have demonstrated the formation of tubulin probes' aggregates in our previous study published in 2018 (ref.1). We have shown the correlation between fluorogenicity and aggregation in the same study. In fact, our data also demonstrated a good correlation between cLogP (spiro lactone) and aggregation, but not between cLogP (zwitterion) and aggregation (see FigureR1 below). It is important to note that aggregate formation is continuous process and it is influenced by multiple processes. We have attempted to perform dynamic light scattering experiments in the past, but results were strongly dependent on the buffer conditions and the starting concentrations of the probes. In summary, taking all observations from current and previous studies, we strongly believe that cLogP is the best parameter which can be used to estimate the probe aggregation behavior. Importantly, cLogP of the probes are following the same trend as cLogP of the corresponding free dyes (see FigureR1d). Thus, we have included cLogP of the dyes in the Table 3 and edited the main text accordingly.

Figure R1. Estimation of the probes' aggregation. **a.** The principle of assay used to estimate the aggregation extent. Correlation between the tubulin probe's aggregation and cLogP of its spirolactone (**b**) or cLogP of its zwitterion (**c**) forms. Data taken from ref. 1. The data points are fitted to equation: $y = 10^{(\text{Intercept} + \text{Slope} \cdot x)} + \text{offset}$. **d.** Linear correlation between cLogP of the dye's spirolactone and the corresponding CTX probe. cLogP was calculated using ChemDraw 21.0.0 software.

Reference

- 1 Lukinavičius, G. et al. Fluorescent dyes and probes for super-resolution microscopy of microtubules and tracheoles in living cells and tissues. *Chem Sci* 9, 3324-3334, doi:10.1039/c7sc05334g (2018).